# A modular model integrating metabolism, growth, and cell cycle predicts that fermentation is required to modulate cell size in yeast populations

Marco Vanoni[1,2☉]*, Pasquale Palumbo[1,2☉], Federico Papa[1,3], Stefano Busti[1,2], Laura Gotti[1,2], Meike Wortel[4], Bas Teusink[5], Ivan Orlandi[1,2], Alex Pessina[1,2], Cristina Airoldi[1,2], Luca Brambilla[1,2], Marina Vai[1,2], Lilia Alberghina[1,2]*

**1** SYSBIO Centre for Systems Biology, Milan, Italy, **2** Department of Biotechnology and Biosciences, University of Milano-Bicocca, Milan, Italy, **3** IASI-CNR, Rome, Italy, **4** University of Amsterdam, Swammerdam Institute for Life Sciences, Amsterdam, The Netherlands, **5** Vrije Universiteit Amsterdam, Systems Biology Lab, AIMMS, Amsterdam, The Netherlands

☉ These authors contributed equally to this work.
* marco.vanoni@unimib.it (MV); lilia.alberghina@unimib.it (LA)

## Abstract

For unicellular organisms, the reproduction rate and growth are crucial fitness determinants and functional manifestations of the organism genotype. Using the budding yeast *Saccharomyces cerevisiae* as a model organism, we integrated metabolism, which provides energy and building blocks for growth, with cell mass growth and cell cycle progression into a low-granularity, multiscale (from cell to population) computational model. This model predicted that cells with constitutive respiration do not modulate cell size according to the growth conditions. We experimentally validated the model predictions using mutants with defects in the upper part of glycolysis or glucose transport. Plugging in molecular details of cellular subsystems allowed us to refine predictions from the cellular to the molecular level. Our hybrid multiscale modeling approach provides a framework for structuring molecular knowledge and predicting cell phenotypes under various genetic and environmental conditions.

## Author summary

The budding yeast *Saccharomyces cerevisiae* is a widely used single-celled experimental organism used to study fundamental biological functions, such as metabolism, cell growth, and cell division. We used the budding yeast to create a simple computational model that connects metabolism (which provides energy and materials for growth) with how the cell grows and goes through its cycle from birth to cell division. Simulation of the computational model quantitatively describes relevant properties of large yeast populations growing on different food sources.

**Data availability statement:** The code to run single cells or populations of iMeGrocy-2, as well as of Hy-iMeGroCy-2 can be found at https://github.com/FedePapa83/Simulation-codes.git. Users will find a README file as well as 2 zipped folders named "iMeGroCy" and "Hy-iMeGroCy." These folders contain the codes for the numerical simulations of the populations of the integrated models iMeGroCy-2 (the coarse-grained model) and Hy-iMeGroCy-2 (the Hybrid iMeGroCy-2 with the G1/S transition molecular plug-in). Each folder contains a tutorial file explaining how to run the programs and where the numerical outputs can be retrieved. DataSet used in this study is available on the same repository (https://github.com/FedePapa83/Dataset) as two separate Excel files (one for main text figures, one for supplementary).

**Funding:** This work was supported by grants to the ISBE-SYSBIO infrastructure to LA and MV. The study also received financial support from the Italian Ministry of University and Research (MIUR) through the grant "Dipartimenti di Eccellenza - 2017" to the University of Milano-Bicocca, Department of Biotechnology and Biosciences, coordinated by MV, and from "ELIXIRxNextGenerationIT" (Code IR0000010)-CUP B53C22001800006 to MV. F. P. and S. B. were partially supported by ISBE-SYSBIO. The funders had no role in study design, data collection and analysis, decision to publish, or preparation of the manuscript.

**Competing interests:** The authors have declared that no competing interests exist.

Analysis of the model predicted that cells relying on constant respiration do not change their average size based on growth conditions. We confirmed these predictions by experimental analysis of yeast mutants with defects in early glycolysis or glucose transport. By substituting elements of the simple model with detailed molecular information, we improved our predictions from the cellular to the molecular level. This hybrid modeling approach helps organize molecular data and predict how cells behave under different genetic and environmental conditions.

## Introduction

The coordination of metabolism, cell growth, and the cell cycle is a central property of all proliferating cells. The genetic, molecular, metabolic, and cell physiological understanding of the budding yeast *Saccharomyces cerevisiae* is unsurpassed by those of other organisms [1,2]. Therefore, budding yeast is a convenient model organism for studying this coordination since many yeast genes involved in essential cellular functions are evolutionarily conserved up to humans, making yeast an attractive experimental model not only in cell, molecular, and systems biology but also for human cancer and drug discovery [3].

Specific phases of the cell division cycle impose distinct metabolic demands, particularly for energy production and biosynthesis of macromolecular precursors. Nutrient availability influences cell cycle progression; for instance, carbon or nitrogen depletion leads to $G_0/G_1$ arrest [4,5], while energy store mobilization at the $G_1/S$ transition is driven by oscillations in cyclin-dependent kinase (Cdk) activity [6–8].

Metabolic fluxes are regulated through multilayered mechanisms, including transcriptional control, post-translational modification, and allosteric regulation of enzymatic activity. These regulatory layers integrate nutrient uptake with ATP production and anabolic processes—particularly RNA and protein synthesis—that drive biomass accumulation and facilitate cell cycle progression [5,9–12]. In nutrient-rich environments, proteome allocation is heavily skewed toward growth-related functions [13].

Signaling pathways such as glucose signaling, the protein kinase A (PKA) and target of rapamycin (TOR) pathways play key roles in integrating metabolic status with cell cycle progression and growth control [5,11,14–16]. Central metabolites—including acetyl-CoA, NADH/NAD$^+$, and AMP/cAMP—serve as intracellular signals that modulate these pathways. Notably, glucose-induced transcriptional responses are largely recapitulated by PKA or Sch9 (yeast PKB) activation [17].

The physiological state of a yeast population reflects a dynamic balance between cell growth and division [18]. During balanced exponential growth, both temporal parameters (e.g., doubling time, DNA replication timing, and cell cycle phase distribution) and biochemical parameters (e.g., cell size and protein content) remain stable [19]. In chemostat experiments, populations of exponentially growing yeast cells present carbon source-specific growth-regulated genes that control mitochondrial function, peroxisomes, and synthesis of vitamins and cofactors [20]. Under these conditions, cells adopt larger sizes and grow faster at higher dilution rates, where

metabolism is primarily fermentative [21–26]. The preference for fermentation over respiration in fast-growing cells is also widespread in higher eukaryotes, including cancer cells [27–29].

Yeast cells reproduce asymmetrically by budding, yielding progressively larger parents and smaller daughters at each division (Fig 1A). To maintain size homeostasis, daughters spend more time in the $G_1$ phase than their cognate parent cells before reaching the critical size required to enter a new cell cycle [12,31]. Thus, each division round produces increasingly larger parents and daughters with shorter division times, giving rise to wide heterogeneity in the structure of yeast cell populations [32].

The protein distribution (i.e., the frequency of cells in a population with a given protein content), measured *via* FITC staining and flow cytometry, provides a quantitative proxy for cellular growth and division status. It is closely linked to the age distribution, which relates to the "actual" state of population growth and depends on the growth law of an individual cell, i.e., the law of protein accumulation during a cell cycle. Protein distributions indicate how a population is growing at the time of analysis and thus provide an accurate snapshot of the current physiological state of a yeast population [23]. For instance, protein distributions differ in cell populations that are exponentially growing on different carbon sources [32] or as cells approach and reach the stationary phase due to glucose exhaustion and later resume growth by using ethanol as a carbon source [30]. Experimental distributions differ from theoretical ones due to biological variability and instrumental noise, which must be considered in modeling efforts. When considering biological and instrumental noise, these theoretical distributions change in shape and closely resemble experimental distributions (Fig 1B; [30]).

In line with the criteria proposed in [33], we extend a previously developed coarse-grained model (*iMeGroCy*) that integrates metabolism, ribosome biogenesis, protein synthesis, and cell cycle progression in an average yeast cell [34]. The updated model (*iMeGroCy-2*) incorporates population structure, enabling direct comparison with experimental protein distributions obtained by flow cytometry, used as the principal biological readout to study yeast populations growing

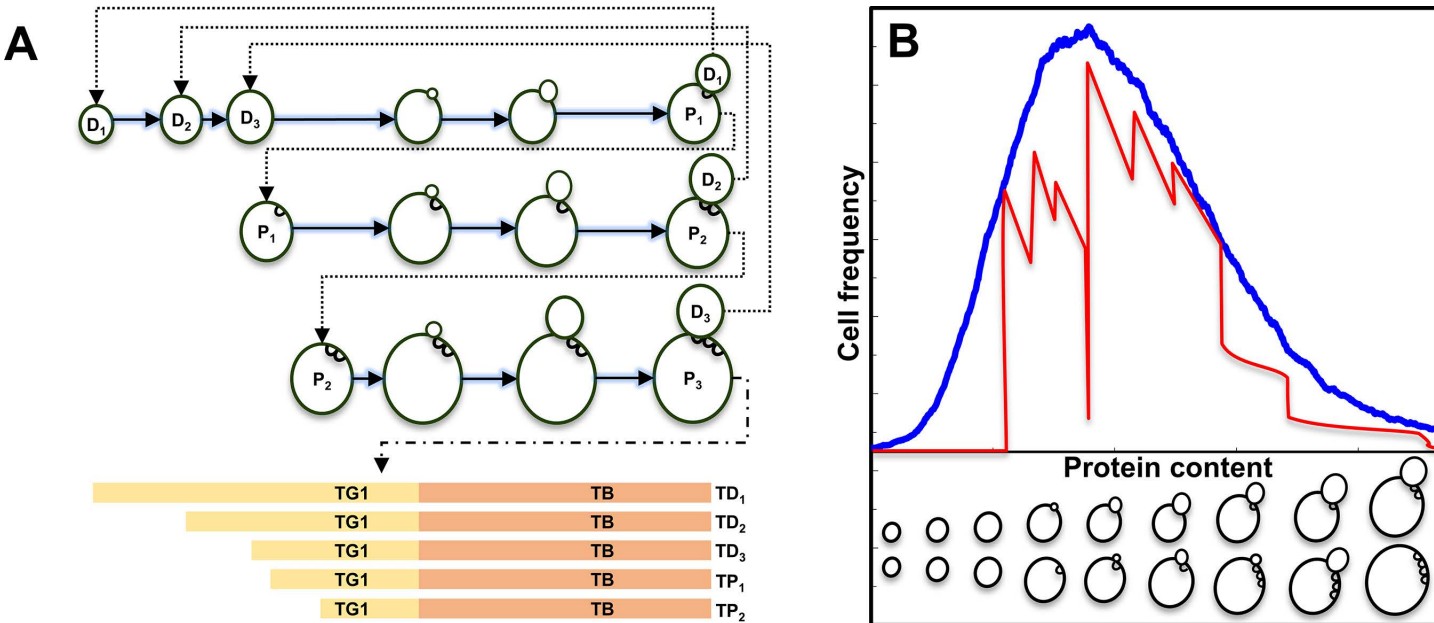

**Fig 1. Cell cycle and cell size in yeast. (A)** Schematic view of the cell cycle for parents (P) and daughters (D) with different genealogical ages. $T_{G1}$ and $T_B$ are the duration of the unbudded ($G_1$) and budded ($S+G_2+M$) phases, respectively. **(B)** Size distribution of a yeast cell population. The red line is the theoretical intracellular protein content density function (adapted from [30]). The blue line is the fitting of experimental protein content distribution (obtained by flow cytometry) with "protein density functions with variability" originated by theoretical function. The lower strip schematically depicts how cells of different genealogical ages but with similar size distribute along the size profile.

in different media and with different genotypes. Additionally, the model now includes ethanol metabolism, broadening its applicability to different nutrient conditions. By adjusting a limited set of input parameters, the model reproduces key physiological metrics—such as protein content distributions and temporal growth parameters—under varying environmental and genetic conditions. This enables the identification of specific cellular processes affected by perturbations. We experimentally validate the model prediction that the type of glucose metabolism —specifically, the balance between fermentation and respiration—drives the nutritional modulation of cell size in exponentially growing yeast populations, and discuss the results in terms of protein allocation and the interplay between growth and cell cycle parameters. Finally, we provide proof that the model's modular, top-down architecture enables its integration with more detailed molecular sub-models, proposing *iMeGroCy-2* as a framework for the stepwise construction of a comprehensive whole-cell model of *S. cerevisiae*, capable of linking gene-level information to population-level phenotypes.

## Results

### General structure and novel functionalities of the *iMeGroCy-2* model

*iMeGroCy-2* (Fig 2) is a significant evolution of the previously published *iMeGroCy* model [34]. In summary, the integrated model identifies three functions: *(i) Metabolism, (ii) Cell growth, and (iii) Cell cycle and division.* The interconnection of two modules, a Metabolism & Growth (*MeGro-2*) module and a Growth & Cycle (*GroCy-2*) module, allows the modeling of the complete biological cycle of yeast populations (Fig 2).

(i)    *Metabolism*: Through a resource allocation algorithm, *MeGro-2* feeds parameters (the desired RNA/protein ratio and the rate of protein synthesis per ribosome) to a dynamic module (*GroCy-2*). *MeGro-2* includes lumped glycolytic and gluconeogenic pathways, a fermentation pathway, and a respiratory pathway, fed by either pyruvate or ethanol, a glucose transporter unit, bidirectional ethanol transport, and ribosomes devoted to biomass production. Ethanol derived from the outside can be respired or used in a simplified gluconeogenic pathway. *MeGro-2* converts the available carbon source into energy (ATP) and macromolecules and uses a growth optimization algorithm to provide growth-related parameters to the *GroCy-2* module. Therefore, *MeGro-2* acts as a parameter generator, receiving two inputs for each growth condition: the external carbon source concentration ($c_{glc_{ex}}$ or $c_{EtOH_{ex}}$) and the fermentation ratio $F$, calculated from the experimental ethanol/glucose yield $Y_{EtOH}$ for glucose-grown cells or set to 0 for ethanol-grown cells. Using these inputs, *MeGro-2* maximizes the exponential growth rate λ, accurately predicting it in different conditions that use as input different glucose concentrations (and their corresponding ethanol/glucose yield) or ethanol (Fig B in S1 Text).

(ii)   *Cell growth (RNA and protein accumulation)* and *(iii) Cell cycle and division*: In *GroCy-2,* growth and cell cycle parameters not directly provided by *MeGro-2* are directly input by the user (see the S1 Text for the details on how *iMeGroCy-2* works and how its parameter values are set, Section 5). Cell growth is modeled by an Ordinary Differential Equation (ODE) system describing ribosome and protein dynamics (synthesis and degradation). A cyclin-mediated switch triggers the cell cycle, modeled as a series of consecutive timers. Timers $T_{1a}$, $T_{1b}$, and $T_2$ encompass the unbudded, $G_1$ phase. $T_B$ corresponds to the budded phase, which includes the S, $G_2$+M, and $G_{1*}$ phases, of which the latter corresponds to the period that separates nuclear and cell division. The end of the last timer triggers cell division. The *GroCy-2* module is simulated for each cell. Starting from one or more founder cells, the population structure is dynamically constructed over the course of the simulation (Fig 2). Cell-to-cell noise is introduced at cell division in newborn cells, giving rise to a simulated yeast population whose output temporal and cell growth parameters (notably protein distributions) can be directly compared with the equivalent parameters of experimental populations. Fig F panel E in S1 Text presents a prototypical protein distribution generated by *iMeGroCy-2* for the case of 2% glucose in comparison with its corresponding experimental distribution. S1 Text presents a detailed description of the integrated *iMeGroCy-2* model.

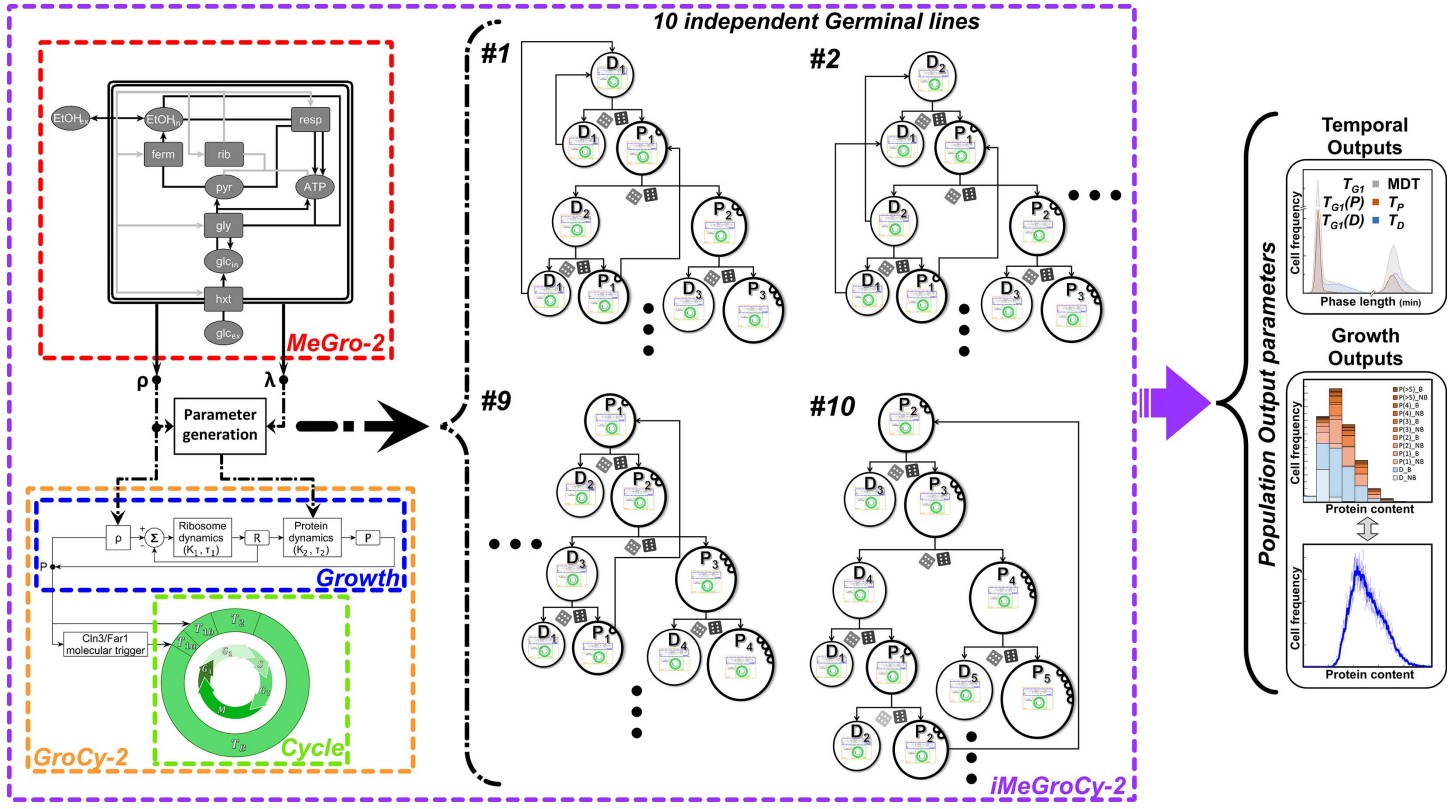

**Fig 2. Structure of the integrated Metabolism *Gro*wth and Cycle (*iMeGroCy-2*) model.** The *iMeGroCy-2* model (purple box) and its modules' inter-connections are depicted, including *MeGro-2* (red box) and *GroCy-2* (orange box), which incorporates the *Growth* (blue box) and *Cycle* (green box) mod-ules. Black and grey arrows inside *MeGro-2* box denote metabolic conversions and protein synthesis, respectively, while rectangles and ovals denote classes of proteins and metabolites, respectively. The *GroCy-2* module simulates a population of several thousand cells, starting from 10 germinal lines, each originating from 10 distinct single progenitor cells. Parent (P) and daughter (D) cells of varying genealogical ages (as defined by their subscript numbers) are illustrated here in the pedigree map. Similar to real-life populations, these cells are linked by a Parent-Daughter connection. As each cell is simulated according to *GroCy-2*, represented by the *GroCy-2* thumbnail inside the cell, the population structure forms as an emergent property of the overall population. The dice symbolize the "random noise" introduced at each division event in newborn cells. *iMeGroCy-2* yields population temporal and growth parameters. An example plot showing the temporal outputs for a 50k population of daughter and parent cells is here reported. The growth outputs histogram plot shows how *iMeGroCy-2* groups together cells of different genealogical ages but similar sizes (i.e., protein content) within a popu-lation. A typical cell size distribution plot is also displayed (blue line graph).

Tables 1 and 2 show why all computational simulations (and their comparison with experimental data) presented in this manuscript *are feasible only* with the new version of the integrated model. Notably, the population structure that emerges from the *GroCy-2* simulation *is essential* for analyzing and extracting biological information from protein distri-butions (see Introduction for more details) obtained during the exponential growth of yeast populations in liquid media.

**Table 1. MeGro-2-vs-MeGro differences.**

| Feature | *MeGro* [34] | *MeGro-2* |
|---|---|---|
| Usable Carbon source | Glucose only | Glucose<br>Ethanol<br>tunable to other carbon sources |
| Gluconeogenesis | No | Yes |
| Respiration | Only from pyruvate | From both pyruvate and ethanol |

**Table 2. iMeGroCy-2-vs-iMeGroCy applications.**

| Feature | *iMeGroCy* [34] | *iMeGroCy-2* |
|---|---|---|
| Single cell | Yes | Yes |
| Population-level | No: only small cell clusters can be analyzed | Yes, allowing direct comparison of simulation results with protein distributions obtained by flow cytometry and temporal cell cycle parameters of populations in balanced or perturbed exponential growth |
| Mutant analysis | Yes, in principle, but no direct comparison with readily determinable population features is possible | Yes, allowing direct comparison of simulation results with protein distributions obtained by flow cytometry and temporal cell cycle parameters of populations in balanced or perturbed exponential growth and their comparison with the experimental and simulated wild type results |
| Open to plug-ins | Yes, in principle, but no direct comparison with readily determinable population features is possible | Yes, allowing direct comparison of simulation results with protein distributions obtained by flow cytometry and temporal cell cycle parameters of populations in balanced or perturbed exponential growth, while keeping the parameter values and simulation results of the coarse-grained model as a reference |

This method of growing yeast cells encompasses not only the growth of shaken flasks but also growth in fermenters and chemostats, which are significant growth conditions in both basic science and applied fields, as yeast that produces valuable products, such as metabolites or proteins, is cultivated in this manner [35]. The population structure is also a prerequisite for using the model as a host for molecularly detailed modules, whose behavior can be studied within a context in which the general population dynamics are retained and act as constraints on the plugged-in module. Finally, only the revised *MeGro-2* supports growth in ethanol (see Fig A in S1 Text and accompanying text for a detailed comparison of *MeGro* and *MeGro-2*).

### *iMegroCy-2* robustly describes the structure of exponentially growing yeast populations.

To evaluate the robustness and biological significance of the identified parameter set, we first assessed the effect of each input parameter on the output growth and temporal parameters. Using the glucose 2% parameter set as a reference set, we conducted a sensitivity analysis by changing a single input parameter at a time while keeping the remaining input parameters constant. The monitored growth outputs are the average value of the protein distribution (<P>), its standard deviation (SD(P)), and coefficient of variation (CV(P)). The monitored temporal outputs are the mass duplication time of the population (MDT) and the length of the $G_1$ phase of parent and daughter cells ($T_{G1D}$ and $T_{G1P}$, respectively). Each input parameter has been moved "forward" and "backward" compared to its nominal value. For each parameter setting (resulting from each parameter variation), the exponential growth of the yeast population was simulated, linking the *GroCy-2* input parameters to the quantitative features of the cell population. S1 Text, Subsection 5.d. provides details on the procedure. Fig 3A–3C report an example of the sensitivity analysis for input parameter $T_2$. Figs H-K in S1 Text show the complete results of the sensitivity analysis summarized as a heat map in Fig 3D. The heat map shows that the perturbation of many input parameters has limited or no effect on the tested outputs, indicating that the model is robust against parameter alterations. At the same time, the map provides suggestions on which parameters could be worth changing to obtain a simulated cell population more closely resembling an experimental yeast population perturbed by chemical or genetic means, thereby hinting at the cellular functions altered by the perturbation (see later chapters for examples). Therefore, the parameterization of the two *iMeGroCy-2* modules, *MeGro-2* (with 27 input parameters, see also Table C in S1 Text) and *GroCy*-2 (with 27 input parameters, see also Table C in S1 Text), is initially dependent on biological constraints and can be subsequently fine-tuned through sensitivity analysis. Together, these steps ensure biological plausibility and robustness of the chosen parameter set under each environmental and genetic condition. See S1 Text (Section 5a) and the Discussion for further information and in depth discussion.

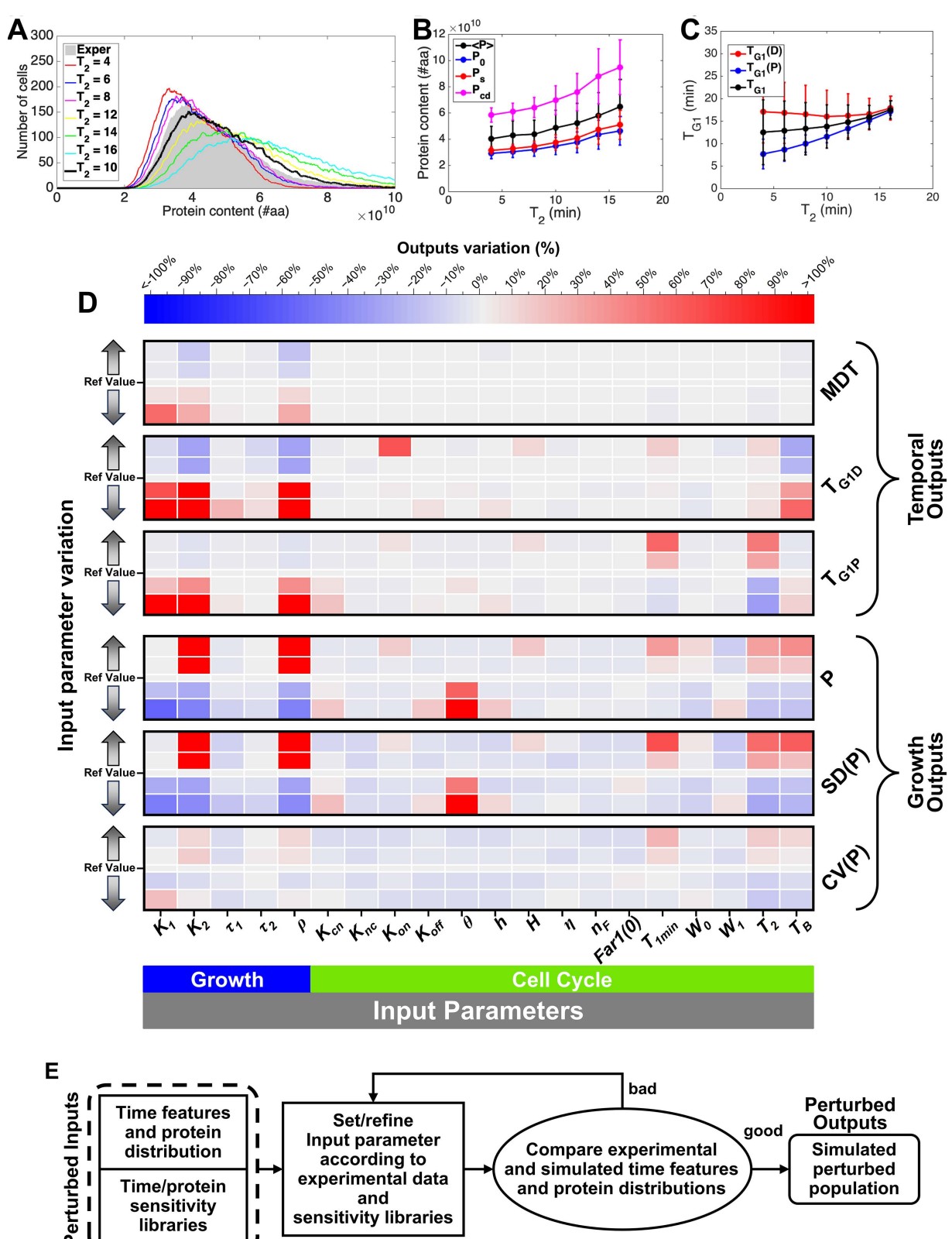

Fig 3. **Robustness and sensitivity analysis of the integrated *iMeGroCy-2* model. (A-C)** Sensitivity analysis with respect to $T_2$ (glucose 2%). **(A)**: simulated protein distributions drawn for different values of $T_2$ and compared to the experimental one related to 2% glucose. **(B)**: protein features

(average and initial cell protein content, critical cell size and size at division) extracted from the simulated populations of panel A, endowed with their standard deviations. **(C)**: $G_1$ phase length for the whole population and for the subpopulations of daughters and parents, related to the simulated populations of panel A, endowed with their standard deviations. **(D)**: Heat map describing the alterations in major output growth and temporal parameters obtained by changing the input parameters of the *GroCy-2* module. The considered temporal parameters are the mass duplication time of the population (MDT) and the $G_1$ phase lengths of parent ($T_{G1Par}$) and daughter cells ($T_{G1Dau}$). The growth parameters are the average value **(P)**, standard deviation (SD(P)) and coefficient of variation (CV(P)) of the cellular protein content. **(E)** Flow chart of the identification procedure for mutational and environmental perturbations.

## Metabolism drives the coordination between cell growth and division

Different nutrients and growth conditions affect the cell size and cell cycle parameters, including the growth rate, budding index, and length of the budded phase (Table N, panel A in S1 Text) [24,36–38]. Yeast cells can use ethanol only through respiration, whereas both respiration or fermentation, which produce ethanol as a byproduct, can metabolize glucose. To investigate the link between metabolism and cell size modulation, we measured ethanol production as a function of glucose consumption (Fig 4A). The lower the glucose concentration in the growth medium, the lower the ethanol produced for each unit of consumed glucose, indicating increased respiratory utilization of sugar (Fig 4B).

As outlined above, modification of only a few parameters may produce yeast populations with significantly different properties. Successful prediction of the outputs of cell cultures grown in various growth media would indicate that the model can correctly identify the crucial parameters affected by the experimental conditions (or genetic mutation; see the following subsections).

First, we proved that our model could simulate different nutritional conditions using roughly the same parameters. As a first step, we run *MeGro-2,* varying its inputs (i.e., different glucose or ethanol concentrations and the fermentative-to-respiratory ratio), keeping all internal model parameters fixed at all nutritional conditions. Indeed, a change in the values of $K_2$ and ρ alone (i.e., the *MeGro-2*-produced outputs) satisfactorily reproduces the output features of yeast populations grown in 5% and 0.5% glucose (Fig 4C-4E). For lower glucose concentrations, this approach captures the reduction in overall dimension and the decrease in growth rate. Still, it leaves space for further improvements, as shown for protein distributions in Fig D in S1 Text. As explained in S1 Text, a minimum of 2 and a maximum of 7 parameters (out of a total of 31) required minor adjustments from the values of the 2% condition. These changes were suggested mainly by experimental data (Table N, panel A in S1 Text) and literature (see Subsection 5b in S1 Text for the details) and allowed the final distributions and corresponding outputs to be obtained, as reported in Fig 4 and Table G in S1 Text.

Fig 4C-4I show the experimental (red) and simulated (blue) protein distributions for populations of the wild-type CEN. PK strain proliferating in media supplemented with either ethanol or different amounts of glucose (from 0.05% to 5%) as the carbon source. The reported data include the average ± standard deviations of the experimental (red) and computational (blue) cell numbers with a given protein content. Despite minor differences between the observed and simulated distributions (drawn in the upper part of panels C-I using the same Y-scale as the protein distributions), the simulated distributions closely matched the experimental distributions. Most differences were confined within one standard deviation, strengthening the ability of the model to faithfully reproduce distributions of cell protein content (used here as a proxy of cell size) of yeast populations in exponential growth [4,39]. It is worth noting that we aim to reproduce *in silico* the qualitative behavior of the population: for instance, by reducing the richness of the nutrient, both experimental and simulated populations are shifted to the left (i.e., cells with a smaller size), and, at the same time, they both change their shape towards a distribution with a variance with a smaller value. These qualitative changes compared to the reference nutritional conditions (glucose 2%) are shared by both experimental and simulated data. This fact can be appreciated by comparing these distributions with the gray distribution (the experimental protein distribution of cells grown in the reference condition, i.e., 2% glucose), providing immediate visual feedback on the ability of the simulated distribution to capture the nutritional modulation of cell size (protein content). Accordingly, the mean, median, mode, and standard deviation

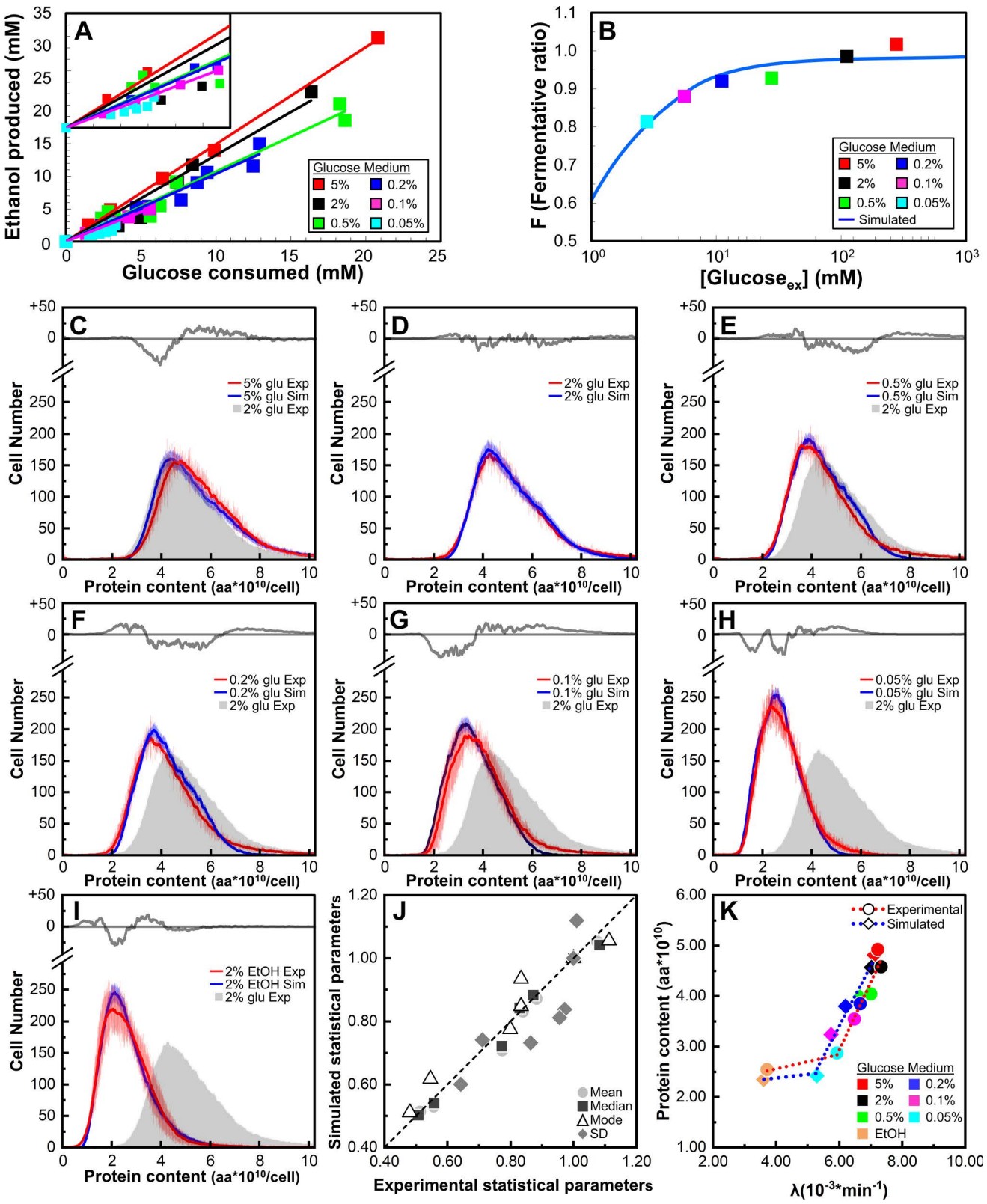

**Fig 4. *iMeGroCy-2* captures the nutritional modulation of cell size and cycle parameters of yeast populations.** (A) Glucose consumption and ethanol production in the wild-type strain. Growth media were sampled during exponential growth to measure glucose consumption and ethanol production. As glucose concentration decreases, experimental points cluster in the left part of the graph, showing heavy superposition. The insert is a magnification of the left portion of the graph. (B)The fermentative ratio F is plotted as a function of the external glucose concentration in the growth medium. Colored dots represent experimental data points measured at different glucose concentrations (ranging from 0.05% to 5% glucose); the continuous blue line represents the best fit of the experimental data based on the saturating function reported by Eq.(S31) and described in the S1 Text, Section 5. (C-I) Comparison of the experimental (red lines) and simulated (blue lines) protein distributions for wild-type cells grown in media supplemented with different glucose concentrations. Mean ± standard deviation (SD) values are reported. In each panel, the gray line in the upper insert represents the difference between the experimental and simulated values. The experimental distribution of cells grown in 2% glucose (filled gray) is also reported as a reference. (J) Correlation between protein distribution parameters of simulated and experimental yeast populations. Every marker series (circles, squares, triangles, diamonds) refers to seven nutrient conditions: (i) EtOH 2%, (ii) glc 0.05%, (iii) glc 0.1%, (iv) glc 0.2%, (v) glc 0.5%, (vi) glc 2%, (vii) glc 5%, ordered from left to right. Each reported statistical value was normalized vs the related (experimental or simulated) 2% glc value. (K)Growth rate ($\lambda$, min$^{-1}$) vs. average protein contents (P, aa) in media supplemented with different glucose concentrations or ethanol. Experimental (red dotted line, circles) and simulated (blue dotted line, diamonds) mean values are reported.

calculated on the corresponding experimental and simulated distributions showed good correlations (Fig 4J). Finally, Fig 4K reports the average experimental (red) and simulated (blue) protein content as a function of the growth rate ($\lambda$, min$^{-1}$). For both experimental and computational distributions, glucose concentration strongly modulates the average protein content, which shows a steep linear correlation with the growth rate $\lambda$ (Fig 4K). Conversely, growth in ethanol-supplemented media significantly decreases $\lambda$, which nearly halves from 0.05% glucose to ethanol (Fig 4K and Table L, panel A in S1 Text), but results in only a further marginal reduction in P (ca. 10%). In conclusion, our results indicate that by reducing the glucose concentration in the growth medium, metabolism progressively shifts toward respiration and that *iMeGroCy-2* captures the change in growth rate and the overall size distribution in exponentially growing populations. As outlined in the Introduction, the protein (size) distribution is a fingerprint of each condition of balanced exponential growth, and the ability to capture this modulation by tweaking only a few parameters indicates that the simplicity of the model includes the most relevant biological interconnection among metabolism, growth, and the cell cycle.

## Metabolism-driven modulation of cell size

In glucose-limited continuous cultures, cell size has been consistently shown to be small and relatively constant at low growth rates when only respiration is active. In contrast, it increases steeply at faster growth rates when cells switch to fermentation, leading to ethanol production [24,36]. Furthermore, cell size increases even at low growth rates if the cells are forced to shift from respiratory to fermentative metabolism [24]. Data presented in Figs 4 and Fig Q and S in S1 Text, and Table A in S1 Text confirmed that glucose can modulate cell size in a dose-dependent manner as a function of the growth rate: increasingly higher sugar concentrations in the growth medium support faster growth rates and larger sizes (Fig Q and Table N, panel A in S1 Text) [10,24,37,38]. The linear correlation between the yield of ethanol and the protein content of the population of yeast cells grown at different glucose concentrations is consistent with the notion that the type of energy metabolism (respiratory, respiro-fermentative, fermentative) influences both cell growth rate and size (Fig S, panel E in S1 Text). We first tested whether the outputs of the metabolism-related module MeGro-2 were consistent with *in vivo* proteome allocations reported in published datasets [40] for yeast cells grown under comparable conditions: namely, 2% glucose and fully respiratory, non-fermentable carbon sources (pyruvate, acetate, glycerol). Proteins were assigned to five classes according to their physiological functions ("glycolysis", "fermentation", "*HXTs*" (glucose carriers), "respiration", "ribosome/translation") and the relative abundance of each group was calculated from experimental data (Fig 5A, right side; see also Fig R in S1 Text and S1 Data). Although acetate, glycerol, and pyruvate (and ethanol, not used in [40]) are metabolized via partially different metabolic pathways, they all converge on mitochondrial respiration, ultimately sustaining ATP production. Therefore, the values for the "fully respiratory" condition in Fig 5 represent the average of the allocation values obtained for acetate, glycerol, and pyruvate. The left part of Fig 5A shows the *MeGro*-2-predicted

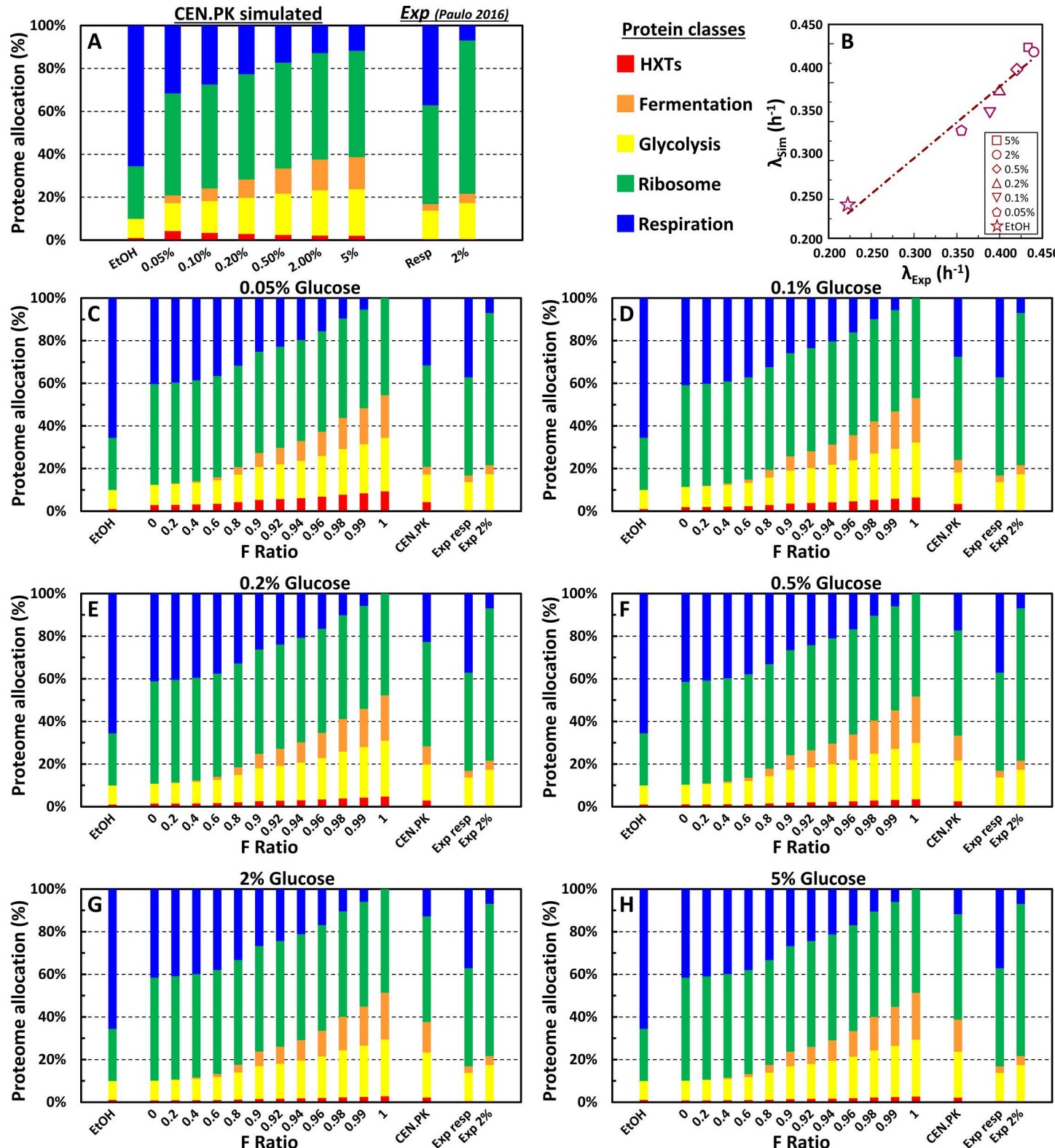

**Fig 5. Proteome allocation and growth rate predicted by *MeGro-2*. (A)**Proteome allocation for the CEN.PK strain under various nutritional conditions simulated by *MeGro-2*. As indicated, the model considers five protein classes (HXTs (glucose carriers), glycolysis, fermentation, respiration, and ribosome/translation), which *in vivo* account for approximately 40% of the entire proteome (according to YeastGenome.org [41]). The experimental

values reported were derived from literature ( [40]: see Fig R in S1 Text and S1 Data; only the five aforementioned classes were considered for the final computation). Since allocation data for ethanol growth conditions were not available, the average value of three non-fermentable carbon sources (glycerol, acetate, and pyruvate) is shown here. **(B)**Correlation between the experimental λ growth rate values and values simulated by *MeGro-2*. **(C-H)** *MeGro-2*-simulated proteome allocations under different fermentative conditions (F ratio between [0, 1]) in various growth media (0.05% to 5% glucose). In each panel, we reported as reference the simulated values for the CEN.PK strain and the experimental literature data (same as panel **A**).

proteome allocations for the same experimental conditions described in Fig 4. *MeGro-2* predictions for both standard (2% glucose) and fully respiratory conditions (*i.e.,* ethanol) align well with the experimental trends: compared with the reference 2% glucose condition, the respiratory pathway investment significantly rises under fully respiratory conditions, whereas translation-related proteins strongly decreases and glycolytic/fermentation enzymes show a moderate reduction (Fig 5A). *MeGro-2* predicts a steady, smooth increase in respiratory allocation from 0.5% to 0.05% glucose and a sharp rise for ethanol (full respiration). The increased abundance in respiratory enzymes is accompanied by a parallel decrease in the fraction of fermentation enzymes. *MeGro-2* predicts a reduction in glycolytic enzymes only for the lowest glucose concentrations (< 0.2%) and (above all) for ethanol (Fig 5A). Moreover, exclusively for ethanol, *MeGro-2* also predicts a significantly reduced allocation in ribosomal synthesis machinery (Fig 5A), which matches the dramatic change in growth rate observed when moving from 0.05% glucose to ethanol, both *in silico* and *in vivo* (Figs 4K and R in S1 Text and Table N, panel A in S1 Text). The close agreement between *MeGro-2*-simulated- and experimental growth rate values is confirmed in Fig 5B.

Next, we analyzed protein allocation as a function of the fermentative ratio (F) across glucose concentrations (Fig 5C–5H). The allocation changes from full respiration (F = 0) to full fermentation (F = 1) closely mirror those in Panel 5A, suggesting that shifts in protein allocation are primarily driven by the F-ratio, rather than by external glucose itself.

While *MeGro-2* does not explicitly model biomass accumulation (*i.e.,* RNA and protein synthesis) and cell division, *GroCy-2* incorporates both functions. Fig 6 summarizes the *GroCy-2* simulation outputs for growth and temporal parameters obtained by systematically varying the F-ratio at different glucose concentrations, *i.e.,* the same conditions analyzed in Fig 5 for the *MeGro-2* outputs. Panel 6A reports the average protein content of the population. The dose-dependent modulation of cell size by external glucose (estimated by the average protein level of the population, P) is substantially diminished for F values ≤0.8, with protein content remaining uniformly low regardless of the external glucose concentration. Similar heatmap profiles appear with both the Standard Deviation (SD(P): Fig 6B) and the Coefficient of Variation (CV(P), i.e., SD(P)/P: Fig 6C). Both values remain constant until glucose metabolism is significantly shifted toward fermentation.

Temporal features, including Mean Doubling Time (MDT: Fig 6D) and the $G_1$-phase length of both parent (Fig 6E) and daughter cells (Fig 6F), are also influenced by the F-ratio. Even under enforced respiration, only minimal modulation of the temporal parameters is observed at F ≤ 0.6, becoming more pronounced at higher F values. When respiration prevails, MDT and $G_1$ length increase significantly in low-glucose conditions, while for F > 0.6, MDT correlates with glucose levels.

The simulation analysis described above predicts that nutritionally dependent modulation of the protein content occurs only when glucose metabolism is strongly shifted toward fermentation, being abolished when cells are "forced" to adopt a respiratory metabolism. To experimentally validate these predictions, we analyzed the behavior of two mutants defective in glucose metabolism: the TM6* strain, which exhibits strongly reduced sugar uptake capacity (Fig S, panel I in S1 Text) [42], and the *hxk2 hxk1* double mutant strain, which lacks two of the phosphorylating isoenzymes that catalyze the first glycolytic step (Fig S, panel J in S1 Text) [43,44]. Cell size parameters (*P*, $P_s$, and $P_0$) were monitored during balanced growth in media supplemented with high (2%) or low (0.1%) glucose levels. Growth in ethanol was used as a reference for fully oxidative metabolism.

In contrast to their wild-type isogenic counterparts, both mutants primarily adopted respiratory metabolism to meet their ATP demands under all tested growth conditions, as suggested by their constitutively high basal and maximal oxygen

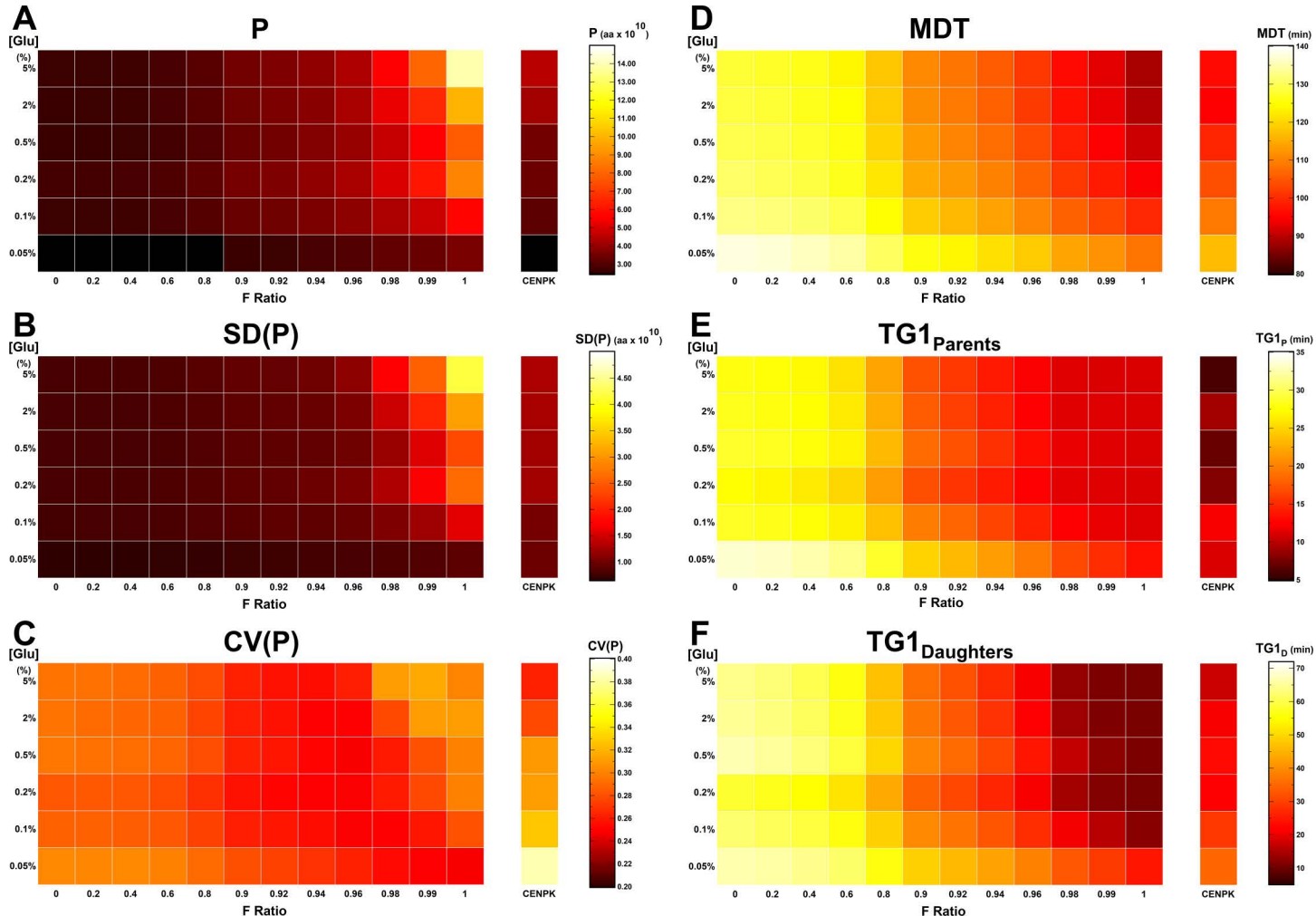

**Fig 6. Modulation of population growth and temporal parameters under various fermentative conditions across different growth media.** *GroCy-2* was used to simulate the growth of yeast populations under various fermentative conditions (F ratio ranging from [0, 1]) across different growth media (spanning from 0.05% to 5% glucose). The model input parameters were set according to the values for the wild-type CEN.PK strain reported in Table E in S1 Text. All output parameter values are displayed as heatmaps. The "CEN.PK" column on the right side of each heatmap presents the simulated output parameter values obtained from *iMeGroCy-2* by setting the F ratios equal to the experimentally measured values for the CEN.PK strain cultivated in the respective growth medium. **(A-C)** Growth parameters: P, SD(P) and (CV(P)) of the intracellular protein content. **(D-E)** Temporal parameters: MDT, $T_{G1P}$ and $T_{G1D}$.

consumption rates ($J_R$ and $J_{MAX}$, respectively: Fig 7A) and the concomitant decrease in F ratio values (Fig 7B) relative to their isogenic wild-type counterparts. This behavior is likely the result of reduced glycolytic flux or relaxation of their glucose-repression mechanisms [42,43]. Contrary to the isogenic wild-type strain and as predicted by our model, the two mutant strains showed a small size regardless of the available carbon source (Fig 7C, red bars), resulting in nearly superimposable protein distribution profiles (Fig 7E-7G). Therefore, the carbon source-dependent modulation of cell size of the two constitutive respiratory mutant strains was substantially compromised in both the TM6* and *hxk2 hxk1* mutants ($P_{2\%Glu}/P_{EtOH}$ = 1.33 and 1.15, respectively). Since nutritional modulation of cell size can be considered fully operative when the protein content ratio during growth in glucose *vs.* ethanol is approximately 1.8 and lost when this ratio is close to 1.0,

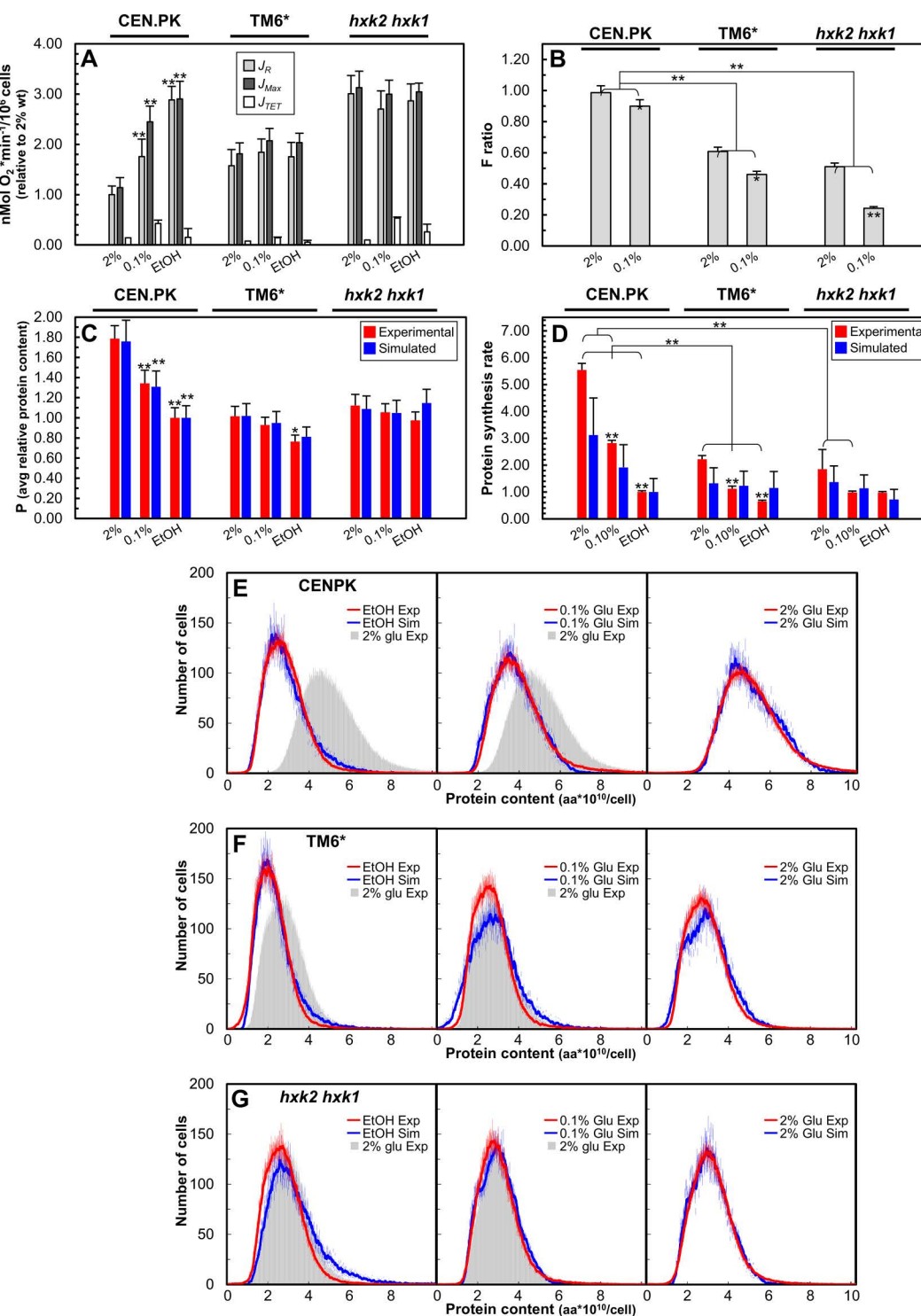

**Fig 7. *iMeGroCy-2* captures metabolism-driven modulation of cell size.** (A) Respiratory parameters determined for the wild-type CEN. PK strain and TM6* and *hxk2 hxk1* respiratory-deficient mutants. Basal respiration rate ($J_R$), uncoupled respiration rate ($J_{Max}$), and non-phosphorylating respiration ($J_{TET}$) during growth in 2% glucose, 0.1 glucose, and ethanol media are shown. Data are relative to wild-type oxygen consumption in a 2% glucose medium. Values are means ± SDs from three independent biological replicates. Statistical significance: *p < 0.05, **p < 0.01, Student's t-test (2% glucose was taken as reference condition for each strain). (B)The F ratio was experimentally determined for wild-type and respiratory-deficient mutants under

different growth conditions. Values are means ± SDs from three independent biological replicates. Statistical significance: *p < 0.05, **p < 0.01, Student's t-test. (C)Experimental (red bars) and simulated (blue bars) average protein contents in the wild-type and respiratory-deficient mutants under different growth conditions. Values are means ± SDs from three independent biological replicates. Statistical significance: *p < 0.05, **p < 0.01, Student's t-test. (E-G)Experimental and simulated intracellular protein content distributions in the wild-type and respiratory-deficient strains obtained under different growth conditions. Mean ± standard deviation (SD) values are reported. (A)Experimental (red bars) and simulated (blue bars) protein synthesis rates for wild-type and respiratory-deficient mutants obtained under different growth conditions. Data are relative to the values measured for the wild-type strain cultivated in ethanol medium. Mean + SD values from at least three biological replicates are reported. Statistical significance: *p < 0.05, **p < 0.01, Student's t-test.

these experimental data reinforce the prediction of our model that fermentation ability is required for nutritional modulation of cell size.

To further explore this aspect, we simulated populations of the TM6* and *hxk2 hxk1* double mutant strains under three nutritional conditions: 2% and 0.1% glucose and 2% ethanol. To this end, we reduced in *MeGro-2* the catalytic coefficient of the flux $v_{gly}$ (namely, $k_{cat,gly}$) by a factor of 1/2 to reproduce the *hxk2 hxk1* low efficiency of glycolysis, and we reduced the catalytic coefficient of the flux $v_{hxt}$ (namely, $k_{cat,hxt}$) by a factor of 5 to account for the TM6* low efficiency of glucose transport (see the S1 Text, Subsections 6.a, 6.b).

Fig 7E-7G indicates that the simulations correctly reproduce the poor—if any—ability of the TM6* and *hxk2 hxk1* mutant strains to nutritionally modulate cell size. In addition, our model correctly predicts a significant reduction in the protein synthesis rate (relative to the rate in the wild-type strain grown in a 2% glucose medium) for both these mutants and for wild-type cells grown in 0.1% glucose and ethanol media (Figs 7D; T, panel A in S1 Text).

**The integrated model describes the growth properties of *whi5* cell cycle mutants: from cellular to molecular analysis**

The *iMeGroCy-2* model integrates three functions. As described in the previous chapter, Fig 7 shows that tuning the *MeGro-2* parameters of mutants in metabolism suffices to reproduce their phenotypic properties. Since most parameters are inherited from the wild-type population, successful simulation of the mutant populations effectively pinpoints altered cellular functions in the mutants. Here, we test the ability of *iMeGroCy-2* to reproduce both the temporal parameters and protein distributions of two mutants in the *WHI5* gene [45–49], which encodes an inhibitor of SBF, one of the key transcription factors required for the $G_1$/S transition. Phosphorylation at multiple sites dissociates Whi5 from SBF, allowing the transcriptional activation of SBF target genes. Four specific "functional" sites of Whi5 must be phosphorylated to release Whi5 from SBF [47,50]. Cells lacking the *WHI5* gene (*whi5Δ* strain) or expressing a Whi5 protein whose functional phosphorylation sites have been mutated to phospho-mimetic glutamate residues (*whi5^4E* strain) anticipate the entry into the S phase, originating small cells [49].

Here, we exploited the *iMeGroCy-2* sensitivity map to understand what parameter variations could be responsible. Table M in S1 Text shows the input parameters altered for the simulations of the *whi5* mutants, and the *iMeGroCy-2* outputs obtained by simulations. The model predicted a mild reduction in the rate of protein synthesis for *whi5* mutants. Experimental data confirmed that the *whi5* mutants showed a decrease in the protein synthesis rate (Fig 8C).

The model allowed the reproduction of the main features of the protein distributions of the *whi5Δ* and *whi5^4E* mutants, capturing the reduction in the size of exponentially growing populations [45,46,48,50,51] (Fig 8A and 8B), indicating that the mechanism altered in the mutant involves the setting and duration of cell cycle timers.

Because of its structure, *iMeGroCy-2* cannot provide molecular details on the timer alteration in the *whi5* mutants. However, its modular structure may allow easy molecular module integration, allowing the opportunity to refine the model's descriptive and predictive ability from the cellular to the molecular level. As a proof of principle, we replaced the Cln3/Far1 trigger and the periods describing the $G_1$ phase with a dynamic molecular model of the $G_1$/S transition (Fig 8D), which is

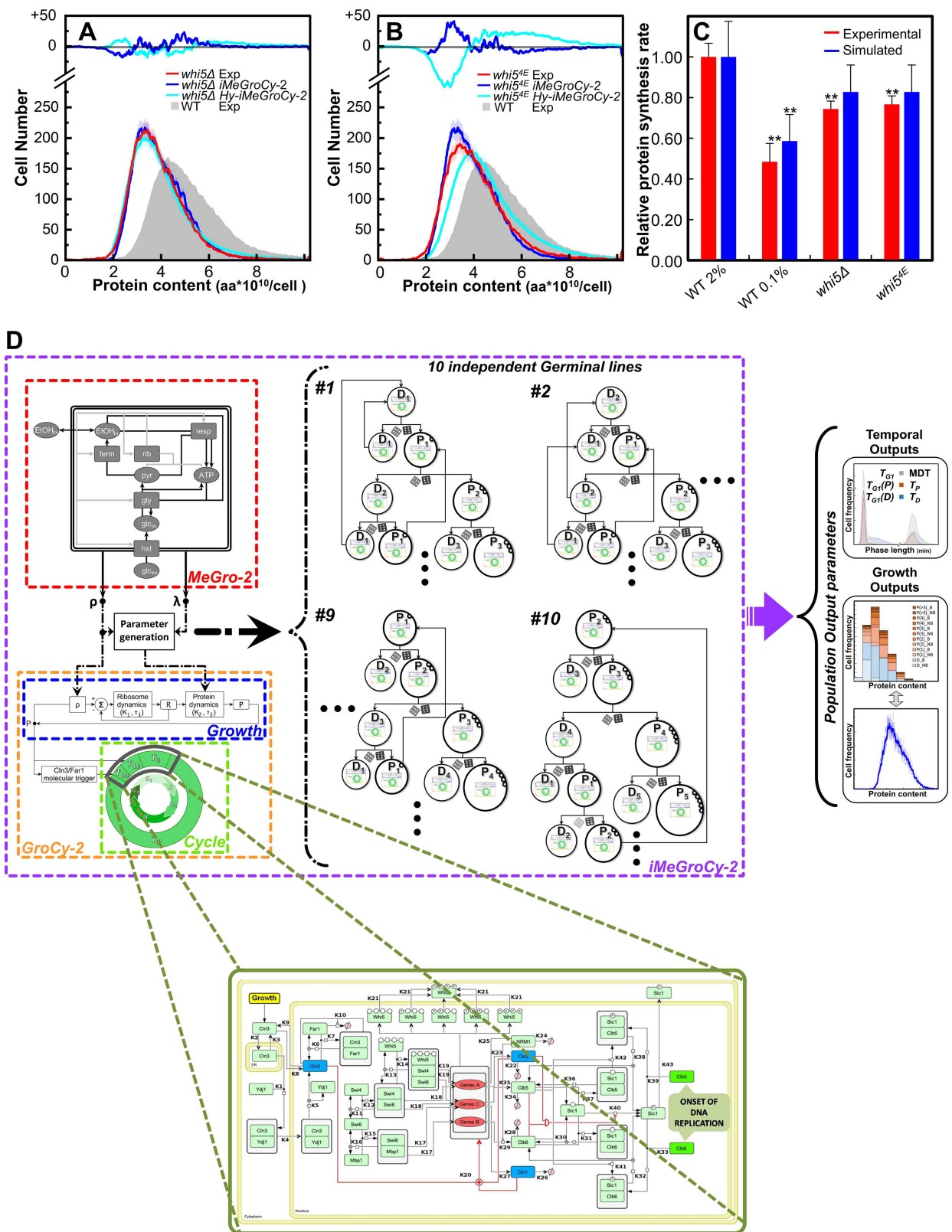

**Fig 8. *iMeGroCy-2* acts as a scaffold for a molecularly detailed module of the G₁/S transition.** (A-B)Protein distributions are shown for mutants *whi5Δ* (A) and *whi5⁴ᴱ* (B) as predicted by the hybrid *Hy-iMeGroCy-2* model (dark blue lines) and by the *iMeGroCy-2* model (cyan lines), compared to

their related experimental distributions (red lines)). Data are Mean ± SD values. In each panel, the cyan and blue lines in the upper insert represent the difference between the experimental and simulated values. The experimental distribution of wild-type cells cultivated in 2% glucose (filled gray) is also reported as reference. Lower inserts show the temporal parameters for both experimental and simulated populations. (C)Experimental (red bars) and simulated (*iMeGroCy-2*, blue bars) protein synthesis rates for different strains and growth conditions. Data are relative to the wild-type strain cultivated in 2% glucose medium. Mean + SD values from at least three biological replicates are reported. Statistical significance: *$p \leq 0.05$ and **$p \leq 0.01$, Student's t-test. (D)The hybrid *Hy-iMeGroCy-2* model, with the $G_1$/S transition molecular module plugged into *iMeGroCy-2* replacing the Timers $T_1 + T_2$.

described in [49]. The resulting hybrid *Hy-iMeGroCy-2* model (see S1 Text, Subsection 5.e for details on the use of such a molecular model as an *iMeGroCy-2* plug-in) quantitatively described the temporal parameters and protein distributions of the wild-type strain at both high (2%) and low (0.05%) glucose concentrations (Fig 8A and 8B). These results are obtained by setting total Whi5 = 0 for the *whi5Δ* mutant; on the other hand, for the simulation of the *whi5^{4E}* mutant, we reduced the binding affinity between Whi5 and SBF by suitably reducing the value of $k_{14}$ to 1/10 of the nominal value in Table J in S1 Text, and we also reduced the clearance rate of Whi5 soon after the binding of SBF/Whi5 (i.e., parameter $\alpha_W$ in [49]) More details can be found in S1 Text, Subsection 6.e. Analysis of the *whi5* mutants (Fig 8A and 8B) showed a good agreement between the experimental and simulated temporal and growth parameters, confirming the predictive power of the hybrid model. It should also be noted that this procedure not only shows the ability of our integrated model to act as a scaffold for molecularly detailed models of selected pathways while retaining the general dynamics of cell functions but also expands the descriptive and predictive ability of the $G_1$/S model that in its published version cannot be directly compared with experimental results obtained on exponentially growing cell populations.

## Discussion

Metabolism, cell growth, and the cell cycle account for most protein investment under various nutritional conditions in yeast [13]. Metabolism extracts energy from nutrients and sustains yeast cell growth, primarily through net RNA and protein synthesis [5,9,10], which regulate the cell cycle [11,12]. In this study, we constructed and validated *iMeGroCy-2*, a coarse-grained, multilevel dynamic model comprising two interconnected modules: *MeGro-2* and *GroCy-2*. Designed as an integrated suite that describes the connection between metabolism, cell growth, and the cell cycle, these modules can also function independently. *MeGro-2* is constrained by the concentration of the available carbon source and its mode of utilization (as indicated by the fermentative ratio F), which can be experimentally determined or set by the user. *MeGro-2* then feeds its outputs to *GroCy-2*, where the growth and cell cycle output characteristics of the individual simulated cells result in populations (wild-type or mutant) according to the pedigree structure intrinsic to the model. *iMeGroCy-2* quantitatively predicts cell protein distributions - a quantitative measure of cellular growth and division status and temporal parameters in response to both genetic variants and carbon source availability.

An emerging trend in cell modeling is moving toward multi-scale integration, where individual cell behavior is modeled with molecular precision, allowing researchers to simulate and predict population-level phenomena with high fidelity. [52] introduced a multi-scale modeling framework that extends their prior whole-cell model of *E. coli* to the colony level. By embedding individual cell models within a shared dynamic environment, the framework captures both intracellular processes and collective colony behavior. Applied to antibiotic resistance -a major public health concern driven in part by phenotypic heterogeneity- the model enabled detailed simulation of *E. coli* responses to tetracycline and ampicillin, two antibiotics with distinct mechanisms of action, thus bridging molecular and population scales. *SimulCell* is an agent-based simulator for eukaryotic cell proliferation, migration, and division on flat substrates, grounded in time-lapse microscopy data [53]. Similar to *SimulCell*, our model captures individual cell proliferation and variability. However, our framework goes beyond mere rule-based approaches, since it leverages a low-granularity whole-cell algorithm, allowing proliferation to emerge from molecular-level interactions. By incorporating higher-resolution molecular models (see later for a fuller discussion), our model offers cell-specific molecular insights comparable to those of [52].

The quantitative outputs of a multi-parameter model—such as the one presented here—that seeks to describe complex biological functions often lack the resolution necessary to identify each parameter within the parameter space uniquely. This limitation arises from a widely recognized intrinsic property of complex biological models known as *sloppiness*, which refers to the highly anisotropic geometry of the model's parameter space [54]. Specifically, model behavior tends to be highly sensitive to perturbations along a limited number of stiff parameter combinations, while remaining largely insensitive to variations along a multitude of sloppy directions.

Identifying stiff and sloppy directions within the parameter space—i.e., combinations of parameters whose variations produce substantial or negligible changes in model outputs—offers more meaningful insights into the model's qualitative dynamics than conventional analysis based on individual parameter variation [54]. The prevalence of sloppy directions renders the task of accurately estimating all parameters from experimental data fundamentally ill-conditioned, particularly in the presence of measurement noise. This does not preclude the utility of such models in generating reliable simulations and predictive outcomes, even in the face of parameter uncertainty. Analogous to biological cells that maintain robust behavior despite stochastic fluctuations in individual molecular reactions, models can remain predictive within certain bounds of parameter variability [55–57].

We conducted a sensitivity analysis to assess the robustness of selected population-level outputs—specifically, temporal dynamics and protein distributions—to individual parameter perturbations in the *GroCy-2* model. The resulting heatmap (Fig 3D) reveals that most perturbations occur in sloppy directions, indicated by minimal output changes (gray regions). Similarly, reproducing wild-type temporal and growth phenotypes under varying glucose or ethanol conditions required adjusting only a small subset of parameters (Fig 4C–4I). Based on the sensitivity analysis (Fig 3D), we also pinpointed key parameters whose variation accounts for the temporal and growth behaviors of mutant strains impaired in one of the model's three core functions: metabolism, growth, or cell cycle (Figs 7, 8 and P in S1 Text). This identification is crucial for linking observed phenotypes to specific biological processes at a resolution consistent with the model's structure.

Parameter sloppiness implies that various parameter combinations can produce similar phenotypes. As proof of concept, Fig M in S1 Text illustrates that different combinations of parameters may yield similar phenotypic outputs when parameters are allowed to co-vary in sloppy directions (see, e.g., the cloud of simulations with varying combinations of parameter values belonging to "Cluster Low," which generate nearly superimposable protein distributions, Fig N in S1 Text). Conversely, there are stiff directions within a reduced set of parameter variations that primarily drive changes in phenotypic properties: for instance, this occurs when increasing growth-related parameters $\rho$ and $K_2$, shifting the output populations towards the "Cluster High," resulting in a dramatic increase in cell size compared to the "Cluster Low."

*MeGro-2* simulations indicate that nutrient-dependent cell size modulation requires glucose fermentation. *MeGro-2* allocates resources across five core biosynthetic processes: glucose transport, glycolysis, fermentation, respiration, and translation—based on a growth-optimizing algorithm (Fig 5). Under predominantly respiratory conditions (F-Ratio $\leq 0.8$), protein allocation remains largely unchanged (Fig 5), and *iMeGroCy-2* predicts modest shifts in temporal dynamics across glucose concentrations (Fig 6). At higher F-Ratios, a significant reallocation from respiratory to fermentative enzymes occurs (Fig 5C–5H). Ribosomal and translational protein levels are stable in glucose but decline in ethanol, consistent with reduced growth rates in ethanol and in 0.05% glucose, a near-fully respiratory condition (Fig 5B).

Under prevailing fermentative conditions (F-Ratio $> 0.8$), the growth rate ($\lambda$) becomes glucose-dependent. Although *iMeGroCy-2* adopts *MeGro-2*-derived parameters, it computes $\lambda$ independently as an emergent property of population-level dynamics. Model predictions correlate closely with experimental measurements (Figs 5B and Fig E in S1 Text). Elevated glucose accelerates growth, advances Cln3 accumulation, and shortens $G_1$ duration. Since the budded phase remains relatively constant, faster-growing cells divide at larger sizes, producing larger newborns and increasing protein distribution heterogeneity, while mean doubling time and $G_1$ duration decline (Fig 6D-6F).

Empirical data show that translation-related proteins in yeast scale linearly with growth rate, whereas glycolytic enzymes and chaperones decrease under respiratory conditions (Figs 5A and R in S1 Text; [40,58–60]). Fast-growing

cells (bacteria, fungi, and mammals) often favor fermentative metabolism instead of respiration to generate ATP under nutrient-rich conditions, even in oxygen-rich environments (aerobic glycolysis: Crabtree and Warburg effect [29]). Rather than "wasteful", such a metabolic strategy is considered a trade-off between energy yield and protein costs to maximize growth [61,62]. In fact, despite the lower ATP yield of fermentation relative to respiration, glycolytic enzymes have been reported to have a superior ATP synthesis capacity per enzyme mass than respiratory enzymes. As a result, fermentation has been proposed to produce energy faster and at significantly lower proteomic costs than respiration [61–69]. Consistent with this view, *MeGro-2* predicts that forcing respiration under nutrient-rich conditions may prevent the optimal allocation of limited proteomic resources, severely limiting the capacity of yeast cells to maximize growth and modulate their size according to the available carbon source. Experimental validation supports model predictions: the TM6* and *hxk1 hxk2* mutants, locked into a mostly respiratory metabolism even in glucose media, exhibited slow growth, reduced protein synthesis and no cell size modulation (Fig 7). Therefore, fermentation in nutrient-rich environments likely facilitates the optimal proteome allocation required for growth and size control.

This view has been challenged by recent findings suggesting that respiration may actually offer superior proteomic efficiency than fermentation, allowing several respiratory yeasts to grow faster and outcompete the glycolytic ones under certain conditions [70]. The study postulate that fermentation may instead have evolved as a robust adaptation to hypoxic environments: since cells are relatively inefficient at metabolic proteome remodelling, adopting a fermentative metabolism that works well under both aerobic and oxygen-limited conditions would be more advantageous. While *MeGro-2* does not impose intrinsic efficiency differences between metabolic modes, it allocates proteome resources optimally based on the availability of carbon sources. *iMeGroCy-2* incorporates these dynamics to generate emergent, system-level behaviors that allow for estimations of protein distributions, average protein content, and intra-population protein content variability.

The loss of carbon source–dependent cell size modulation observed in the TM6* and *hxk2 hxk1* double mutants cannot simply be attributed to their reduced growth rates, which fall within the range where cell size becomes growth rate–independent (Fig S in S1 Text) [24,36]. In fact, TM6* cells grown in 2% glucose were significantly smaller than expected based on their growth rate and comparable in size to wild-type cells cultured in 0.2% glucose (Fig S in S1 Text). Moreover, the *snf3 rgt2 gpr1 gpa2* mutant, which lacks extracellular glucose sensing, still shows a (slightly altered) nutritional modulation of cell size, despite a growth rate reduction similar to that observed for the TM6* mutant (Fig S in S1 Text, $P_{2\%Glu}/P_{EtOH}$ = 1.506; Table N, panel B in S1 Text). Notably, this sensing mutant exhibited a significantly higher ethanol yield than TM6* and *hxk2 hxk1* mutants (Fig S and Table O in S1 Text), suggesting a metabolic shift toward fermentation (a model-predicted fundamental prerequisite for glucose-dependent cell size modulation, Fig 6A). Nonetheless, in contrast to the wild-type strain, the ethanol yield values for this sensing mutant were relatively constant at all tested glucose concentrations, likely due to its impaired sugar uptake system (Fig S in S1 Text, panel I). Taken together, these findings suggest that during steady-state growth, cell size is primarily set by metabolic mode, with nutrient sensing acting at a higher regulatory level. This conclusion is further strengthened by a recent study showing that in many yeast species the rate of glucose metabolism sets cell size, but not *vice versa* [26].

A whole-cell computational model is designed to simulate the complete behavior of a living cell by integrating the complex interactions among cellular components like proteins, metabolites, genes, and regulatory networks, aiming to offer an integrated, systems-level perspective of cellular functions, such as metabolism, signaling, and gene expression (reviewed in [33]). To date, only a handful of whole-cell models are available, since developing and curating these kinds of models is challenging and labor-intensive [71–73]. The first whole-cell model was developed for *Mycoplasma genitalium* a human urogenital parasite whose biological functions were modelled by 28 interacting modules [74]. A fully dynamic, 3D spatial whole-cell model of a synthetic minimal bacterium with a reduced genome of 493 genes, a number close to the 525 genes of *M. genitalium*, has also been published [75]. Several refined versions of the original whole-cell model of *E. coli* [76], whose genome contains more than 4000 genes (https://ecocyc.org/ECOLI/organism-summary) have been developed [77–79]. The current model includes 43% of known gene products that collectively simulate key cellular processes

(including metabolism, DNA replication, transcription and translation [77,78]) and has been recently extended, as mentioned above, to a population-scale, producing a "whole-colony" model [52].

The granularity level of any model critically affects the questions that can be asked. Constraint-based models can describe the salient features of yeast growth, which is, however, modeled simply as an increase in total biomass without accounting for macromolecular syntheses [80,81]. The genome-scale metabolic model (GEM) of yeast metabolism has been continuously refined and enhanced since the initial publication [82], culminating in the recently published *Yeast9* [83]. The integration of GEMs with COnstraint-Based Reconstruction and Analysis (COBRA) methods, including Flux Balance Analysis (FBA), has yielded extensive insights into the core structure and functional dynamics of metabolic pathways [72,80,84]. GEMs can also be constrained by incorporating enzyme reaction thermodynamics ($k_{cat}$), post-genomic data (including transcriptomics, proteomics, metabolomics and fluxomics), subcellular compartmentalization and energetic costs of biological processes (see for instance [80,81,85–87]), significantly enhancing the predictive capabilities of classical GEMs.

In principle, Genome-scale metabolic models (GEMs) may also provide a solid basis for building increasingly complex multiscale models by incorporating modules of different cellular processes [33,69,85]: the development of the WM_S288C computational model is an important first step towards a comprehensive whole-cell model of yeast, that can be used to decipher the complex relations between genotype and phenotype [88], although significant challenges remain unsolved [33,69,81].

While established whole-cell models adopt a bottom-up approach, reconstructing cellular complexity from individual components to achieve high-resolution descriptions, *iMeGroCy-2* employs a top-down strategy. It models the core functions of eukaryotic cells—metabolism, growth, and the cell cycle—at low granularity, functionally interlinked to capture emergent cellular behaviors.

The modular design of *iMeGroCy-2* enables substitution of coarse-grained components with increasingly detailed molecular modules. Each major module – or part of it - can be replaced or refined into interconnected submodules, preserving functional behavior and facilitating efficient parameter tuning. This plug-in framework ensures simulation consistency between high- and low-resolution modules and allows resolution choice based on the biological question under investigation. To validate this scaffold approach, we integrated an existing molecular model of the $G_1$/S transition [49] into the Cell Cycle module. The resulting hybrid model (Fig 8D) accurately replicated growth dynamics, timing, and protein distributions for both wild-type and mutant strains, matching results obtained with the coarse-grained model (Figs 8A, 8B, Fig O in S1 Text). While both versions predict decreased protein content and synthesis rates in mutants, the hybrid model offers mechanistic insights by identifying specific molecular alterations. For example, simulations of *whi5* deletion or its constitutively phosphorylated mutant (*whi5$^{4E}$*) reproduced expected phenotypes. While the coarse-grained model can only suggest which macro-functions are affected in mutants, the hybrid molecular model enables detailed analysis of the specific molecular players involved. Simulations of either *whi5* or *whi5$^{4E}$* mutations can be performed independently, expanding the model's utility. In the case of *whi5$^{4E}$*, the model not only captures the expected shift in protein distribution toward lower values, but also suggests an alteration in the interaction constants between SBF and Whi5$^{4E}$. Refinement of these parameters—pending experimental determination—would enhance the model's accuracy and predictive capability.

These results confirm the model's utility as a scaffold for integrating mechanistic modules, capable of capturing mutant phenotypes and guiding further refinements. Foreseeable expansions of the integrated model could include both the description of cell growth and metabolism. A model of ribosome assembly – a highly complex process involving more than 200 proteins and assembly factors [89] could replace the simple equations currently describing ribosome biosynthesis. In this context, the coarse-grained model's prediction of impaired protein synthesis in the *rsa1*-null mutant (Fig P in S1 Text) suggests the involvement of Rsa1 in a limiting step of ribosome biogenesis, leading to a severe reduction of the protein synthesis rate [90]. Incorporating detailed models of ribosome assembly could validate these findings and reveal regulatory interactions unresolvable at lower resolutions.

Constraint-based models can be integrated with ODE-based frameworks, enabling reformulation of the allocation problem based on measured proteomic constraints. Recent work has demonstrated genome-scale resource allocation

optimization in *E. coli* [91] and applied similar strategies to yeast central metabolism under the assumption of coupled flux and protein levels [64,69]. Such an approach would allow for effective integration of (multi-) omic data, as discussed above. As shown for *Mycoplasma genitalium* [74] and more recently for *E. coli* [76,78] and a living minimal cell [75], constraint-based models can be integrated with ODE-based frameworks, enabling reformulation of the allocation problem based on measured proteomic constraints.

Although a full list of molecular models describing specific yeast cellular functions (reviewed in [33] pluggable in *iMeGroCy-2* goes beyond the scope of this paper, it is worth citing models simulating processes like cells cycle [92], DNA replication [93,94], transcription [95] and translation [96]. The online availability of a Yeast Cell Model Data Base (YCMDB, https://www.tbp-klipp.science/ycmdb/) may be instrumental in collecting and providing curated, high quality data for model testing and validation.

In conclusion, our coarse-grained algorithm functionally integrates the core macro-processes of proliferating cells—metabolism, growth, and the cell cycle—each governed by thousands of gene products. The sensitivity analysis (Figs 3D and Fig H–K in S1 Text) enables the identification of macro-functions disrupted by environmental or genetic perturbations. The model's modular structure supports the stepwise integration of detailed molecular modules and subsystem models, whose development can be enhanced through AI-based methodologies [97]. This allows mechanistic models to be quantitatively evaluated within a validated framework reflecting the genetic and environmental influences on yeast populations. Crucially, the model bridges a key gap by linking molecular mechanisms to population-level behaviors—growth, cell cycle progression, and environmental responses—features often absent in isolated molecular models. Given the evolutionary conservation of core proliferative mechanisms from yeast to humans, this framework holds promise for application in mammalian systems. It offers a foundation for understanding deregulated growth in cancer and developing targeted therapies.

Despite advances, current models fall short in capturing the full complexity of living cells. This underscores the need for more integrated approaches that combine omics data with computational models. The emerging concept of Artificial Intelligence Virtual Cells (AIVCs) proposes a promising avenue, leveraging AI to create comprehensive digital twins of cellular functions [98,99].

## Methods

### Yeast strains and plasmids

All strains used in this study (Table 3) were derivatives of *S. cerevisiae* W303-1A [100] and CEN.PK2-1C (Euroscarf Cat# 30000A). Cassettes for *WHI5* and *RSA1* gene deletions were generated via PCR by using genomic DNA of specific haploid strains from the deletion collection (Euroscarf) as template and short primers (~20 bp long, sequence available upon

**Table 3. *List of yeast strains used in this study.***

| Strain | Relevant genotype | Reference |
|---|---|---|
| W303-1A | *MATa leu2–3,112 ura3–1 trp1–1 his3–11,15 ade2–1 can1–100 GAL SUC mal* | [100] |
| *rsa1* | W303-1A *rsa1Δ::KANMX* | This study |
| CEN.PK2-1C | (*MATa ura3–52 trp1–289 leu2–3,112 his3Δ1 MAL2-8C SUC2*) | Euroscarf |
| *snf3 rgt2 gpr1 gpa2* | (CEN.PK2-1C *snf3Δ::loxP rgt2Δ::KANMX gpa2Δ::LEU2 gpr1Δ::his5$^{Sp}$*) | |
| *whi5* | (CEN.PK2-1C *whi5Δ::KANMX*) | [49] |
| *whi5$^{4E}$* | (CEN.PK2-1C *whi5::KANMX whi5$^{4E}$::URA3*) | [49] |
| TM6* | CEN.PK2-1C *hxt17Δ ura3–52 gal2Δ::loxP stl1Δ::loxP agt1Δ::loxP ydl247wΔ::loxP yjr160cΔ::loxP hxt13Δ::loxP hxt15Δ::loxP hxt16Δ::loxP hxt14Δ::loxP hxt12Δ::loxP hxt9Δ::loxP hxt11Δ::loxP hxt10Δ::loxP hxt8Δ::loxP hxt514Δ::loxP hxt2Δ::loxP hxt367Δ::loxP; HXT7prom-TM6\*-HXT7term ura3–52::URA3* | [42] |
| *hxk2 hxk1* | CEN.PK2-1C *hxk2Δ::loxP hxk1Δ::HIS3* | This study |

request) designed at least 100 bp upstream/downstream of the ATG and stop codon using the Perlprimer software ([http://perlprimer.sourceforge.net/](http://perlprimer.sourceforge.net/) [101], as described [102]. Alternatively, we generated deletion mutants by the PCR-mediated gene disruption method (short flanking homology *loxP::marker::loxP/Cre* recombinase system [103]. Cells were transformed by the standard lithium acetate procedure [104]. Deletion of the targeted gene sequences in transformants was confirmed by colony PCR. Whenever needed, the markers used for gene disruptions were removed by inducing the recombination of the flanking *loxP* sequences through the Cre recombinase expression [103]. For the inactivation of *HXK1*, an *hxk1::HIS3* disruption cassette was amplified by PCR using genomic DNA from a *hxk1*-null strain as a template [105]. for sugar sensing in establishing the catabolite-repressed state}.

gpa2 null strains were constructed by one-step gene replacement using a *gpa2::LEU2* disruption cassette obtained by digestion with *PstI* of the pUC19-*gpa2::LEU2* plasmid [106].

A DNA fragment encoding the *whi5^4E* mutant was synthesized *de novo* by Eurofins ([www.eurofins.com](www.eurofins.com)) and sub-cloned into the YIplac211 integrative plasmid under the *WHI5–545_pr* native promoter, yielding the construct YIplac211–*545_pr*-*WHI5^4E*. Single-copy genomic integration of the construct at the *URA3* locus was verified by quantitative PCR as described [49].

## Determination of growth parameters

Yeast cultures were grown in synthetic complete minimal medium, containing 0.67% (w/v) yeast nitrogen base (YNB), appropriate quantities of the 'drop-out' amino acid–nucleotide mixture, and supplemented with either 2% (w/v) glucose, 0.05% (w/v) glucose or 2% (v/v) ethanol.

The growth of cultures was monitored as the increase in cell number using a Coulter Counter model Z2 (Beckman Coulter). Cell size analysis was performed using a Coulter Z2 Particle Cell Analyzer (Beckman-Coulter). The budded cells ($F_B$) fraction was scored by direct microscopic observation of at least 400 cells fixed in 3.6% formaldehyde and mildly sonicated.

The fraction of parent budded ($F_{PB}$) and unbudded ($F_{PNB}$), daughters budded ($F_{DB}$) and unbudded ($F_{DNB}$) cells inside the population was determined by bud scar analysis after 10 min staining with Calcofluor white (25 μM in PBS, protected from light): at least 800 cells were scored by direct microscopic counting under a Nikon Eclipse E600 fluorescence microscope, equipped with a 100X, 1.4 oil Plan-Apochromat objective, and standard DAPI filter set. Images were digitally acquired using a Leica DC 350F camera and processed with the FiJi software [107].

The formulas used to calculate the length of the budded period ($T_B$), of the binucleated phase ($T_{G1*}$), the average parent cycle time ($T_P$), and the average daughter cycle time ($T_D$) were [32]

$$T_B = \log_2\left(1 + F_B\right) * T \tag{1}$$

$$T_{G1*} = \log_2\left(1 + F_{G1*}\right) * T \tag{2}$$

$$T_P = \log_2\left(F_B/F_{PB}\right) * T \tag{3}$$

$$T_D = \log_2\left(F_B/F_{DB}\right) * T \tag{4}$$

where $T$ is the overall duplication time of the population, $F_B$ is the percentage of budded cells, and $F_{PB}$ and $F_{DB}$ are the fractions of budded parents and daughters in the whole population, as determined by bud scar analysis. Due to the asymmetrical division, the parent and the daughter cycle times must satisfy the equation

$$e^{-\mu T_D} + e^{-\mu T_P} = 1 \tag{5}$$

where $\mu$ is the growth rate given by

$$\mu = \ln(2) / T \tag{6}$$

## Determination of protein (cell size) and DNA cellular contents

Yeast cell size can be estimated by various methods (see Table 1 in [108]. In this study, we used the cellular protein content (evaluated by flow cytometry) as a proxy of cell size. Flow cytometry analysis gave size results comparable with chemical determination of cellular protein content and volume analysis by Coulter counter (Fig Q in S1 Text).

Samples of growing cultures (at least $2*10^7$ cells) were collected and fixed in 70% ethanol before the cytofluorimetric analysis. For protein staining, cells were washed once with cold PBS (3.3 mM $NaH_2PO_4$, 6.7 mM $Na_2HPO_4$, 127 mM NaCl. 0.2 mM EDTA, pH 7.2), resuspended in 1 ml of freshly prepared protein staining solution (fluorescein isothiocyanate (FITC) 50 μg/ml in 0.5 M $NaHCO_3$) and incubated for 1 h in ice protected from light. After incubation, cells were washed three times with PBS and resuspended in 1 ml PBS.

For DNA staining, cells were washed once with PBS, resuspended in 1 ml of PBS with RNAse 1 mg/ml, and incubated overnight at 37°C. Cells were then washed once with cold PBS resuspended in 1 mL of propidium iodide staining solution (0.046 mM propidium iodide in 0.05 M Tris–HCl, pH=7.7; 15 mM $MgCl_2$) and incubated for 30 min in and in the dark. The RNAse treatment was omitted for RNA staining.

For DNA/Protein bi-parametric analysis, RNA-treated cells were resuspended in 1 mL of a 1000-fold diluted FITC staining solution (50 ng/mL FITC in 0.5 M $NaHCO_3$) and incubated for 30 min in ice protected from light; cells were then washed three times with PBS and resuspended in DNA staining solution [109].

Cell suspensions were sonicated for 30 s before the analysis, which was performed with a FACSCalibur (Becton Dickinson) instrument equipped with an Ion-Argon laser at 488 nm laser emission. The sample flow rate during analysis did not exceed 500–600 cells/s. Typically, 30,000 cells were analyzed for each sample.

The average protein content of the entire population ($P$), at the beginning of the cell cycle ($P_0$) and at the onset of DNA replication ($P_s$) were determined as the average fluorescence intensity of appropriately gated cells from the density plot derived from FACS analysis of double DNA/protein stained cells. Data analysis and gating were performed using Flowing software (http://flowingsoftware.btk.fi/). $P_s$ and $P_{cd}$ (the protein content at cellular division) are correlated population parameters: their ratio $h$ is given by the equation [23]

$$h = P_{cd}/P_s \tag{7}$$

The value of $P_P$ (the average protein content of parent cells) can be calculated from either [23]

$$P_P = P/\mu * (T_P - T_D + T_D * e^{\mu T_P}) \tag{8}$$

or

$$P_s = P_P * e^{\mu(T_P - T_B)} \tag{9}$$

For an immediate comparison between experimental and simulated data, the cellular protein contents of the yeast populations obtained by flow cytometry were converted from "arbitrary fluorescence units/cell" to "number of polymerized amino acids (aa)/cell," according to the procedure described in the paragraph *Comparisons of experimental and simulated distributions.*

For chemical determination of protein content, exponentially growing cells ($5 * 10^8$ total cells) were collected by filtration, washed twice with ice-cold 5% Trichloroacetic Acid (TCA), and stored at − 20 °C for at least 12 h. Samples were then

slowly thawed in ice, resuspended in 5 mL of perchloric acid 0.3 N, and heated for 30 min at 90 °C. After centrifugation (10 min at 3500 rpm), the pellet was resuspended in 1 N NaOH and incubated overnight at room temperature on a rolling drum. Samples were then centrifuged (10 min at 3500 rpm, and the resulting supernatants were used for protein determination according to the microbiuret method, using bovine serum albumin as a standard.

Mean cell volume distribution was acquired with a Coulter Counter Z2 instrument using the AccuComp Z Software (Beckman Coulter). Briefly, samples of cell cultures were fixed with 3.6% formaldehyde, mildly sonicated, and diluted in 10 ml of 0.9% NaCl solution before size analysis.

## Determination of glucose consumption and ethanol production rates

At different time points, samples of culture were collected and centrifugated. Extracellular glucose and ethanol levels in the supernatants were evaluated by $^{1}$H-NMR as previously reported [110]. Alternatively, glucose consumption and ethanol production were evaluated through a Hyperlab automatic multi-parametric analyzer SMART (Steroglass, San Martino in Campo, Perugia, Italy) and specific enzymatic kits for ethanol and glucose detection (Steroglass). The Hyperlab analyzer, managed by the Hyperlab SMART Software, handles samples, reagents, and dilutions reproducibly without human processing. Metabolite concentration (g/L) was automatically calculated from measured absorbance values with appropriate algorithms that used a standard calibration curve as a reference.

## Protein synthesis assay

We quantified the protein synthesis rate with the "Click-iT HPG Alexa Fluor 488 Protein Synthesis Assay Kit" (Thermo Fisher Scientific), with minor adaptations from the manufacturer's protocol. The Kit detects the incorporation into nascent proteins of HPG (L-homopropargylglycine), a methionine analog containing an alkyne moiety, through labeling with a fluorescent azide (Click-iT reaction). Yeast cells were grown overnight at 30 °C in SC methionine-free medium. For each sample, $1*10^{7}$ cells were collected and resuspendend in 100 μl of growth medium supplemented with 50 μM HPG. After 15 min incubation at 25° C, cells were washed once with ice-cold PBS and fixed for 15 minutes at room temperature with 100 μl of 3.7% formaldehyde in PBS. After fixation, cells were washed twice with 1 ml of 3% BSA in PBS, resuspended in 100 μl of permeabilization buffer (0.5% Triton X-100 in PBS), and incubated for 20 min at room temperature. Permeabilized cells were then washed twice with 1 ml of 3% BSA in PBS, and the "Click-iT reaction" was performed: cells were resuspended in 100 μl of Click-iT reaction cocktail (freshly prepared according to the manufacturer's datasheet) and incubated for 30 minutes at room temperature, protected from light. After removal of the reaction cocktail, cells were washed once with 100 μl of rinse buffer and once with 1 ml of PBS. Cells were finally resuspended in 1 ml PBS and mildly sonicated (twice for 15 s) before analysis. The fluorescence arising from incorporated HPG-Alexa Fluor 488 was detected by flow cytometry with a FACScalibur instrument (Becton Dickinson). The mean fluorescence intensity for each strain/growth condition was evaluated from 50000 cells (Fig T in S1 Text) and normalized to the value of the wild-type strain. Samples incubated in the absence of HPG were used as negative controls. If required, protein synthesis was inhibited by pre-incubating cells for 60 min with 100 μg/ml cycloheximide before HPG addition.

## Estimation of Oxygen Consumption Rates

Oxygen consumption in intact cells was measured at 30° C using a "Clark-type" oxygen electrode in a thermostatically controlled chamber (Oxygraph System, Hansatech Instruments), essentially as described [111]. Briefly, 2 mL of cell suspension at a concentration of $~5 \times 10^{6}$/mL was quickly transferred from the flask to the oxygraph chamber. The basal/routine oxygen consumption rate ($J_R$) was determined from the slope of a plot of $O_2$ concentration against time divided by the cell number. Treatment with 37.5 mM Triethyltin bromide (TET, a lipophilic F0F1-ATPase inhibitor; Sigma) allowed the determination of the non-phosphorylating respiration rate due to proton leakage ($J_{TET}$), whereas the maximal (uncoupled)

oxygen consumption rate ($J_{MAX}$) was measured after addition of the protonophore Carbonyl cyanide-*4*-(trifluoromethoxy) phenylhydrazone (FCCP, Sigma) at 10 μM and of saturating amounts of ethanol (100 mM) as respiratory substrate. The addition of 2 mM antimycin A (Sigma) accounted for non-mitochondrial oxygen consumption. All assays were performed at least in biological triplicate. The net respiration (netR) was estimated by subtracting $J_{TET}$ from $J_R$ and used to calculate the net routine control ratio $netR/J_{MAX}$.

**2-NBDG uptake assay**

The glucose uptake rate was estimated by using the fluorescent non-hydrolyzable analog 2-NBDG (2-Deoxy-2-[(7-nitro-2,1,3-benzoxadiazol-4-yl)amino]-D-glucose; Sigma). Exponential phase cells were harvested by centrifugation, washed once with fresh medium (w/o glucose), and resuspended at $1*10^8$ cells/ml in fresh medium (w/o glucose). 2-NBDG (25 mM stock solution in DMSO) was added to 50 uL of cell suspension (final concentration 0.2-0.4 mM). Cells were incubated for 30 min at 30 °C, collected by centrifugation, washed twice with ice-cold PBS, and resuspended in 1 mL PBS. After mild sonication (2 x 15 s), the mean incorporated fluorescence intensity was detected by a FACScalibur instrument (FL1 filter) from 30000 cells and normalized to the value of the wild-type strain grown in 2% glucose medium. Samples incubated without 2-NBDG were used as negative controls.

**Hexokinase assay**

Hexokinase activity was measured as described [112], with minor modifications. Briefly, ~$3*10^8$ exponentially growing cells were harvested by centrifugation and washed once in ice-cold distilled water. Cells were disrupted by glass bead lysis in 40 mM MES-Tris buffer (pH 6.8) containing 2 mM PMSF. The protein concentration of samples was measured by the Bradford method, and 200 mg of crude extract was used for the phosphorylation assay. Samples were supplemented with 20 mM D-glucose, 10 mM ATP, and 5 mM $MgCl_2$ and incubated at 30°C. The amount of glucose remaining at each time point was measured by using a glucose-oxidase-peroxidase reaction kit (Sigma). Glucose phosphorylation activity was evaluated from the total glucose consumed per unit of time.

**Statistical analysis**

Data are reported as means ± SDs from at least three independent experiments. The statistical significance of the measured differences was assessed by a two-sided Student's t-test (*p < 0.05, **p < 0.01).

Simulated data are also reported as means ± SDs; mean values and standard deviations of the growth and temporal features reported in Figs 4, 7, 8, and D, F, H–K, N–P in S1 Text, and Tables G and K in S1 Text are computed based on simulated populations of about 50000 cells, that is the same size of the experimental data. Values referring to daughters and parents are also given in the same tables and figures, considering the related subpopulations, usually containing about 50% of the total cell population (and, in any case, not less than 20000 cells). Mean values and standard deviations of the protein content distributions reported in Figs 4, 7, and 8 are obtained from 20 populations of about 50000 cells.

**Analysis of yeast proteome allocation under various nutritional condition.**

To evaluate the *in vivo* effects of metabolism on proteome allocation, we examined previously published proteomic datasets of budding yeast cultivated on various carbon sources ( [40]). Proteins were classified into 5 groups (included in MeGro-2), according to their physiological functions: namely, "glycolysis", "fermentation", "*HXTs*" (glucose carriers), "respiration", "ribosome/translation" and "others". Ribosome/translational class includes ribosomal proteins, translational factors and assembly factors involved in ribosome biogenesis. "Respiration" includes enzymes involved in the TCA cycle, glyoxylate cycle, gluconeogenesis and OXPHOS (components of the mitochondrial electron transport chain and ATP synthase). For a complete list of all proteins included in each class see S1 Data. According to YeastGenome.org [41] the fraction of

proteome comprising these protein classes is ~40% in 2% glucose medium. In this dataset, this fraction is ~20% in all the tested C-sources. Values are expressed as percentage number of total proteins/cell.

## Comparisons of experimental and simulated distributions

The experimental data exploited to draw the protein distributions of the yeast populations are given in arbitrary fluorescence units (fu): an array of 1024 channels, formally spanning the number of about 50 thousand cells on the range of protein content. Conversely, the protein content of the simulated populations is given as the number of polymerized amino acids (aa). In order to compare the experimental and simulated protein distributions we need to translate the experimental data in terms of polymerized amino acids and we need to design the range and the length of the histogram bins of the simulated distribution in order to best fit the experimental one, according to the following optimization algorithm: 1) choose a bin length and a position of the first of 1024 equally spaced and not overlapped bins in the aa axis; 2) builds up the histogram of the simulation taking from the whole simulated population a sample of the same size of the experimental one that fits the bin-scale computed at step 1); 3) compute the sum of the square residuals between the experimental histogram, placed on the aa-axis at step 1), and the simulated one, built at step 2). Then, steps 1) to 3) are repeated to minimize the displacement between the two distributions.

Once the best-fitting procedure is applied for the WT strain grown in 2% glucose, the optimal values of the length and of the position of the bins are applied (without further modifications) also to the other glucose (out of 2%) and ethanol concentrations, and for the mutants of a given WT in glucose. Similarly, when dealing with mutants in an ethanol environment, the histogram bins are set for WT ethanol and then applied (without further modifications) also to the mutant distribution.

## Software and data set availability

The code to run single cells or populations of *iMeGrocy-2*, as well as of *Hy-iMeGroCy-2* can be found at https://github.com/FedePapa83/Simulation-codes.git.

Users will find a README file as well as 2 zipped folders named "iMeGroCy" and "Hy-iMeGroCy." These folders contain the codes for the numerical simulations of the populations of the integrated models *iMeGroCy-2* (the coarse-grained model) and *Hy-iMeGroCy-2* (the Hybrid *iMeGroCy-2* with the G1/S transition molecular plug-in). Each folder contains a tutorial file explaining how to run the programs and where the numerical outputs can be retrieved.

The minimal data set (comprising experimental data used in this study) can be accessed on the same repository (https://github.com/FedePapa83/Dataset).

## Supporting information

**S1 Text.** **Fig A. Comparison of the *MeGro*-vs-*MeGro-2* building blocks.** Besides being a by-product of the fermentation, ethanol may be a metabolic source for *MeGro-2*, and it can be exchanged with the external environment (in both directions) by means of a suitable membrane transport mechanism. External and internal ethanol, as well as the new ethanol respiration and the gluconeogenic pathway producing pyruvate from ethanol, are highlighted in bold black in *MeGro-2*. **Fig B. Correlation between experimental and *iMeGroCy-2* simulated values for the growth rate $\lambda$ parameter.** Values (obtained under various nutritional conditions) are reported as relative to the 2% glucose growth media, taken as reference. The green bisector line indicates the ideal, perfect correlation between the two data series. **Fig C. Pairs of ($K2$, $\tau_2$) satisfying the constraint of Eq.(S27) for fixed values of ($\lambda,\rho$).** The red markers refer to the values of $\tau_2$ coming from Table D in S1 Text; the black circle refers to the chosen setting. **Fig D. Protein distributions for different low glucose and ethanol concentrations.** Curves drawn for *GroCy-2* model parameters inherited from glucose 2%, without any further tuning than the one provided by the *MeGro-2* interconnection (green curves), with also a further (though minimal) tuning (blue curves), compared to the experimental ones (red curves). The grey shade refers

to the glucose 2% experimental distribution reference. **Fig E. Correlation between *MeGro-2-vs-iMeGroCy-2* growth rate.** Dots are growth rate λ values obtained under different fermentative conditions (F ratio between [0, 1], color-coded) in various growth media (from 0.05% to 5% glucose). **Figure F. Simulated parameters in evolving and steady state populations. A, B)** time course of the average protein content, for fast (glucose 2%, panel A) and slow (glucose 0.05%, panel B) growth conditions. **C, D)** time course of the budding index, for fast (glucose 2%, panel C) and slow (glucose 0.05%, panel D) growth conditions. The convergence rate depends on the kind of population meta-parameters. The black lines show the time behavior of <P> and BI related to the 10 clusters, while the red line depicts their average value. **E)** simulated vs experimental protein distribution of a yeast population growing in 2% glucose. **Fig G. Sensitivity analysis of the *MeGro-2* output λ with respect to *kcat* parameters.** Relative ratios of λ (normalized w.r.t. the values in 2% glucose) as a function the coefficient of variation of $k_{cat,x}$, $x \in Prot2$. **Fig H. Sensitivity analysis with respect to *GroCy-2* parameters related to growth**. **Left panels:** simulated protein distributions drawn for different variations of the selected model parameter, w.r.t. the nominal value of 2% glucose (compared also to the experimental distribution, grey ghost). **Central panels:** protein features (average and initial cell protein content, critical cell size and size at division) extracted from the simulated populations, endowed with their standard deviations. **Right panels:** $G_1$ phase length for the whole population and for the subpopulations of daughters and parents, endowed with their standard deviations. **Fig I. Sensitivity analysis with respect to the parameters of the molecular trigger (pt I).** **Left panels:** simulated protein distributions drawn for different variations of the selected model parameter, w.r.t. the nominal value of 2% glucose (compared also to the experimental distribution, grey ghost). **Central panels:** protein features (average and initial cell protein content, critical cell size and size at division) extracted from the simulated populations, endowed with their standard deviations. **Right panels:** $G_1$ phase length for the whole population and for the subpopulations of daughters and parents, endowed with their standard deviations. **Fig J. Sensitivity analysis with respect to the parameters of the molecular trigger (pt II).** **Left panels:** simulated protein distributions drawn for different variations of the selected model parameter, w.r.t. the nominal value of 2% glucose (compared also to the experimental distribution, grey ghost). **Central panels:** protein features (average and initial cell protein content, critical cell size and size at division) extracted from the simulated populations, endowed with their standard deviations. **Right panels:** $G_1$ phase length for the whole population and for the subpopulations of daughters and parents, endowed with their standard deviations. **Fig K. Sensitivity analysis with respect to the parameters of the timer sub-model**. **Left panels:** simulated protein distributions drawn for different variations of the selected model parameter, w.r.t. the nominal value of 2% glucose (compared also to the experimental distribution, grey ghost). **Central panels:** protein features (average and initial cell protein content, critical cell size and size at division) extracted from the simulated populations, endowed with their standard deviations. **Right panels:** $G_1$ phase length for the whole population and for the subpopulations of daughters and parents, endowed with their standard deviations. **Fig L. Alterations in major output growth and temporal parameters obtained by changing the GroCy-2 population parameters.** The considered temporal parameters are the population mass duplication time (MDT) and the G1 phase lengths of parent (TG1Par) and daughter cells (TG1Dau). The growth parameters are the average (P), standard deviation (SD(P)) and coefficient of variation (CV(P)) of the cellular protein content. **Fig M. Clustering of virtual populations as coming from multi-parametric sensitivity analysis**. Each point refers to a population. Different sets are reported according to different colors. Besides, the 7 simulated nutritional environments are reported, as well as 4 mutants discussed later. **Fig N**. **Protein distributions from Cluster Low.** Comparison of the protein distributions of 4 simulated populations among the 67 ones that have a distance ($L_\infty$ norm) smaller than 5% from the reference population (nominal parameters used for WT 2% glucose) compared to the nominal one: red, blue, magenta distributions have a MSE < 50 while the green one has a MSE > 150 w.r.t. the nominal distribution. **Fig O. The *iMeGroCy-2* coarse-grained model acts as a scaffold for a molecularly detailed module of the G1/S transition.** Protein distributions predicted by *Hy-iMeGroCy-2* model (blue lines), compared to the related experimental distributions (red lines), for wild-type strains grown in 2% (A) and 0.05%

(B) glucose media. Mean±SD values are shown. In each panel, the gray line in the upper insert represents the difference between the experimental and simulated values. The experimental distribution of wild-type cells cultivated in 2% glucose (filled gray) is also reported as a reference. Lower inserts show the temporal parameters for both experimental and simulated populations. **Fig P. Mutant *rsa1*: simulated and experimental distributions and protein synthesis rate. A)** Comparison of experimental (red lines) and simulated (blue lines) protein distributions for the mutants rsa1 grown in glucose 2% media. Mean±SD values are shown. The gray line in the upper insert in each panel is the difference between experimental and simulated values. The experimental distribution of wild-type cells cultivated in 2% glucose (filled gray) is also reported as a reference. The lower inserts show the temporal parameters for both experimental and simulated populations. **B)** Experimental (red bars) and simulated (blue bars) protein synthesis rates. Data are relative to the values measured for the wild-type strain cultivated in 2& glucose medium. Mean+SD values from at least three biological replicates are reported. Statistical significance: *p<0.05, **p<0.01, Student's t-test. **Fig Q. Yeast cell size evaluated by different methods. A)** The Average cellular protein content of cell population (measured by flow cytometry and expressed in Fluorescence Units (FU); **B)** the average protein content/cell (obtained by chemical dosage, expressed in pg/cell); **C)** the mean cell volume (measured by Coulter Counter, in fL). Values were plotted *vs.* the growth rate ($\lambda$, min$^{-1}$) for wild type cells cultivated in media supplemented with different glucose concentrations or ethanol. Inserts in panel A show representative protein distribution profiles, determined by flow cytometry. *Fig R. In vivo yeast proteome allocation during growth on various carbon sources.* Values reported are % total proteins #/cell (data taken from [22]). "non-fermentable" is the average allocation during growth on three carbon sources glycerol, acetate and pyruvate (here also reported). Proteins were manually grouped into 6 classes according to their physiological functions: "glycolysis", "fermentation", "HXTs" (glucose carriers), "respiration", "ribosome/translation" and "others". In this dataset, these classes cover ~20% of the entire proteome in all the tested C-sources. For a complete list of all proteins included in each class see S1 Data. **Fig S*. Metabolism-driven modulation of cell size.* A-D)** Growth rate ($\lambda$, min$^{-1}$) vs. protein contents ($P, P_s, P_0$, Fluorescence Units) in media supplemented with different glucose concentrations or ethanol. **E-H)** protein contents ($P, P_s, P_0$, Fluorescence Units) *vs.* ethanol yield. **I)** Glucose uptake capacity monitored by evaluating incorporation of the fluorescent 2-NBDG glucose analogue. Values are relative to the glucose uptake capacity of wild type cells cultivated in 2% glucose medium. The anomalous results of the *hxk2 hxk1* mutant are likely due to artifacts. **J)** Reduction of glucose kinase activity in the *hxk2 hxk1* mutant. **Fig T. *In vivo* protein synthesis rate.** Fluorescence incorporated in neo-synthesized proteins was measured by cytofluorimetric analysis. Representative distribution profiles from various strains and growth conditions are shown. **Table A. *MeGro-2* molecular players. Table B. *MeGro-2* fluxes. Table C. *MeGro-2* input parameters. Table D. *iMeGroCy-2* population output parameters for pairs ($K_2, \tau_2$) satisfying the constraint of Eq. (S27) for fixed values of ($\lambda, \rho$). Table E. *iMeGroCy-2* input parameters. Table F. *Impact of variability in MDT and protein partitioning at division.* Table G. *iMeGroCy-2* population output parameters. Light cyan rows refer to control experimental data. Table H. Set of multiparametric variations providing *in silico* different nutritional conditions and mutants. Table I. Set of multiparametric variations providing *in silico* populations in Cluster High. Table J. *G1_S module* input parameters. Table K. *Hy-iMeGroCy-2* population output parameters. Light cyan rows refer to control experimental data. Table L. *iMeGroCy-2* input/output parameter variations for mutants *hxk2 hxk1 and* TM6*. Table M. *iMeGroCy-2* input parameter variations for mutants *rsa1, whi5* and *whi54E*. Table N. Panel A. Growth parameters of the CEN.PK wild type strain. Panel B. Growth parameters of various mutants.* Table O. Ethanol yield. Table P. Growth parameters of strains impaired in glucose sensing mechanisms.**
(DOCX)

**S1 Data. In vivo yeast proteome allocation during growth on various carbon sources. Values reported are % total proteins #/cell (calculated from data found in [22]). Proteins were manually grouped into 6 classes according**

to their physiological functions: "glycolysis", "fermentation", "HXTs" (glucose carriers), "respiration", "ribosome/translation", and "others". "non-fermentable" is the average allocation during growth on the three carbon sources glycerol, acetate, and pyruvate.
(XLSX)

## Acknowledgments

The authors are grateful to Prof. E. Boles for the *hxt-null* mutant and to the late Prof. S. Hohmann for the TM6* mutant. The authors wish to thank Emanuele Tallarico for help in code optimization and population simulations.

## Author contributions

**Conceptualization:** Marco Vanoni, Bas Teusink, Lilia Alberghina.

**Investigation:** Marco Vanoni, Pasquale Palumbo, Federico Papa, Stefano Busti, Laura Gotti, Meike Wortel, Ivan Orlandi, Alex Pessina, Cristina Airoldi, Luca Brambilla, Marina Vai.

**Methodology:** Marco Vanoni, Pasquale Palumbo, Federico Papa.

**Software:** Federico Papa, Stefano Busti.

**Visualization:** Federico Papa, Stefano Busti.

**Writing – original draft:** Marco Vanoni, Pasquale Palumbo, Federico Papa, Stefano Busti, Bas Teusink, Lilia Alberghina.

**Writing – review & editing:** Marco Vanoni, Pasquale Palumbo, Federico Papa, Stefano Busti, Lilia Alberghina.

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
