## [Decision Letter · Decision Letter 0]

Dear Prof. Vanoni,

Thank you very much for submitting your manuscript "A modular model integrating metabolism, growth, and cell cycle predicts that fermentation is required to modulate cell size in yeast populations" for consideration at PLOS Computational Biology.

As with all papers reviewed by the journal, your manuscript was reviewed by members of the editorial board and by several independent reviewers. In light of the reviews (below this email), we would like to invite the resubmission of a significantly-revised version that takes into account the reviewers' comments.

In particular, some of the mathematical descriptions of the models come directly from previous publications, raising questions about the originality and novelty of the writing and perhaps the work. The manuscript is challenging to follow, mainly because the model's details are scattered throughout the main text, Methods section, and Supporting Information. Consequently, reviewers found it extremely difficult to understand how the original model is structured and the significance of the updates to the current model over the previously published models. 

We cannot make any decision about publication until we have seen the revised manuscript and your response to the reviewers' comments. Your revised manuscript is also likely to be sent to reviewers for further evaluation.

Sincerely,

William Cannon

Academic Editor

PLOS Computational Biology

Stacey Finley

Section Editor

PLOS Computational Biology

Reviewer's Responses to Questions

**Comments to the Authors:**

Reviewer #1: In this work, Vanoni et al. present a coarse-grain mathematical model linking metabolism, cell growth and cell cycle to predict the regulatory role of fermentation in modulating cell size within yeast populations. The model builds on the author’s previous model (Palombo et al 2018) where MeGro and GroCy modules are presented.

Comments:

1) Considering that the model builds on the author's previous model, it would be good to be clear about the actual novelty of this work. This is also reflected in the mathematical description of the models, which comes directly from previous publications.

2) Many of the simulation results presented indicated that some sort of noise is incorporated into the model. On page 25 under Statistical analysis the Authors refer to some sort of binning procedure. But I wonder if this can actually correctly represent the noise of the system. Can you please explain how the noise is incorporated?

3) The procedure for ‘adjusting the parameters’ sounds like an ad hoc method with no proper quantification of the trustworthiness of the obtained parameters. Instead of the classical trial and error procedure, as the authors name it, it would be better to use more established methods (for example, some Bayesian approaches to automatically tune the enzyme kcat values from proteomics-constrained GEMs) The way that procedure is now done may also be the reason that the parameters are led to the state in which they will show ‘excellent agreement between experimental and computational protein distribution (as indicated on p12 and Fig 6), but there are no quantitative values shown to evaluate the fitness.

Reviewer #2: In this manuscript, the authors update an existing modular model of yeast growth with more detailed modules to make some predictions on how changes in environmental and genetic conditions affect the distribution of cell sizes in yeast. Given the way that the results are currently presented in the manuscript, however, I was very confused as to how the model was built, how their parameters were determined, and what outputs from the model are actual “predictions” as opposed to model outputs that were simply fit to experimental data. While I can see the computational model presented in the manuscript as being potentially useful to the scientific community, I am not entirely persuaded by the points made by the authors in the current version of the manuscript. I would like to see the authors address these suggestions before I can make a decision on this manuscript:

- The paper adds a few updates to an existing model of yeast metabolism, growth, and cell cycle that enables new functionalities. As someone without any prior knowledge of this original model, it was extremely difficult to understand how the original model is structured and the significance of these updates. The authors briefly summarize the changes made to the MeGro module and the structure of the GroCy module in the first few paragraphs of the Results section, but I would suggest starting off with a few paragraphs on just describing the mathematical structures of these modules and how the two modules are connected, some of which the authors get to in later paragraphs of the same section. I understand that the detailed explanation of the original model must be delegated to previous publications / supplementary materials to a certain extent, but readers should not be expected to have any knowledge about the previous version to follow this work just from the main text. I also think somewhere like Figure 2A would be a great place to explicitly highlight which parts of the model were newly added with this update versus which parts of the model existed in the original version.

- To my understanding, all of the heterogeneity in cell sizes between individual cells, which seems to play a key role in later analyses, seems to be coming from randomizing some model parameters in the GroCy model at every cell cycle. From my understanding of single cell heterogeneity, I would expect much of the differences in cell sizes must also come from differences in the growth rate lambda, but it doesn’t seem like the authors are adding any noises to the MeGro module of their model. I understand that mechanistically incorporating all sources of heterogeneity is something that is out of scope for this paper, but given that the “prediction” of cell size distributions seems to be a big selling point of this paper, I don’t think enough attention is being given to how the heterogeneity was achieved. The coefficient of variation the authors use (0.05) for the random sampling of GroCy, for example, should be part of Figure 2D – I would expect that outputs SD(P) and CV(P) are fairly sensitive to these very arbitrary parameters.

- In Figure 3, the authors briefly describe their process of fitting parameters of their GroCy model such that the distributions of cell sizes match the experimental distributions for each media condition. Given the large number of parameters they were able to modify and the simplicity of the distributions, I am inclined to think that there must exist multiple combinations of parameter values that can achieve the same distributions, and that the set of parameters that the authors have chosen are merely one of these many parameter sets. With a very limited discussion on their parameter fitting process, it is difficult to understand how the authors would argue that these parameter sets are the best sets of parameters. Looking at Table S3, it seems that very few of the parameters were adjusted in terms of GroCy inputs that do not come from MeGro. Could the authors provide justification for why they would think those specific parameters would be different depending on the environmental condition?

- I have a similar comment for Figure 4. The authors provide very little discussion on what parameters they adjusted to simulate the two mutants, and how these adjustments were done. I can check Table S5 to see that adjustments were made on particular parameters in the MeGro model, which presumably led to changes in the growth rate, and one parameter from the GroCy model. Are these parameters relevant to the biological consequences of the mutations of these strains? Are there alternative parameter adjustments that could have achieved the same desired outcome? I am assuming that these parameters were chosen to be adjusted because they are relevant to the first steps of glucose metabolism, but this needs to be more explicitly explained. Also, it is unclear from the text how these parameters were refit for these mutations. The authors state that these parameters “comply with the mutant features and the outputs obtained in the simulations”. What mutant features and outputs are these? If these parameters were fit to match the experimental distribution of cell sizes, as was the case in Figure 3, the authors cannot state that their model “predicted” any of the values that are shown in Figures 4E through 4G.

- For Figure 5, again, there needs to be more discussion on which parameters they ended up altering, how the parameters are relevant to the biology of the mutations, and what model outputs they were fitted against.

- The results that the glucose metabolism mutants and the rsa1 mutant, which is defective in ribosome biogenesis, has a significant reduction in protein synthesis rates does seem particularly interesting or valuable as a conclusion that is coming from a systems biology study that uses an integrated genome-wide computational model. Are there any nontrivial or emergent properties that arise from the completed model that are noteworthy?

- I don’t really understand what Figure 6 is trying to convey, except that the authors were able to replace a part of their GroCy module with a much more complicated sub-model and get pretty much the same outputs as the original version. Unless the authors were able to utilize this much complicated sub-model to do new types of analyses that work with the more fine-grained parameters, I don’t see the value in including this work in this manuscript.

- I was able to access the code for the model from the GitHub link that the authors provided, which contained detailed documentation for the general usage of the model, which I greatly appreciated. What I couldn’t find, however, were instructions on how I could reproduce the distributions of cell sizes that are shown in the figures presented in the manuscript. The version of MATLAB the authors used to generate the figures, as well as a clear list of commands that need to be run to reproduce the figures, would ideally be included in the documentation. I would also ask that the different parameter sets that are specific to each condition and mutant be included in a spreadsheet file that can be directly imported into the code, such that users would be able to quickly run simulations for all conditions and mutants presented in the manuscript. With the current version, it seems like users will need to manually replace the values in parameters.txt with the values provided in the supplementary tables, which substantially hurts the general accessibility of the model.

Here are some of the more minor comments I had:

- There’s an unpaired parentheses in the last paragraph of the introduction.

- The Figures are not referenced in the main text in the same order as they are presented (Fig 1AB -> Fig2 -> Fig1CD).

- The color palette used in some of the schematic figures (Figure 2A, 2B, 3A, 6A) are confusing and inconsistent. I would suggest using less colors in these figures overall, and clearly specifying what the different colors mean if they end up being used.

- What is the “Yeast Genome Transcriptional Control” box supposed to represent in Figures 2A and 6A? Is this another module in the model? I don’t remember any mentions of this module in the main text.

- For the heatmap in Figure 2D, the positive and negative directions should have entirely different color profiles (e.g. one in red, one in blue, 0% in white). With the current color profile it’s extremely difficult to see which parameters have positive or negative changes.

Reviewer #3: Summary:

Cell size homeostasis in budding yeast is well-studied under different carbon sources, including fermentable sugars and non-fermentative sources like ethanol. However, few models have integrated metabolism, growth, and cell cycle regulation. This manuscript expands on a previous computational model designed for glucose respiro-fermentation (iMeGroCyc, Ref. 39), which linked these processes. The key innovation of iMeGroCyc 2.0 (Figure 2) is the incorporation of ethanol energy production through respiration in the metabolism-to-growth module (MeGro) and the creation of a computational pipeline to simulate a yeast population (10^5 cells) undergoing balanced growth. This feature is essential for comparing the model with experimental data, such as flow cytometry, and for using these data to train and refine model parameters, as demonstrated in Figures 3-6. A significant finding of the manuscript is that cell size correlates with growth rate during fermentative growth but not during respiratory growth. This relationship is inferred from their fitted model (Fig. 3) and subsequently tested with experiments on fermentation-deficient mutant strains (Fig. 4). The authors further refine their model with ribosome-biogenesis mutations (rsa1) and cell cycle mutations (whi5_Delta and whi5^{4E}) in Figure 5. The final figure demonstrates that mechanistic molecular networks of the G1/S regulatory pathway can be added to the GroCyc module, allowing the Hybrid model (HyG1_S-iMeGroCy 2.0) to replicate cell cycle mutant data presented in Figure 5.

The strength of this new model (iMeGroCyc 2.0) lies in its ability to be directly compared with experimental data, which enables the model to be fitted and trained using flow cytometry data. Additionally, the model was tested and then refined with experimental data from various genetic mutants.

However, I found the manuscript challenging to follow, mainly because the computational model's details are scattered throughout the main text, Methods section, and Supporting Information. The average PLoS Computational Biology reader would likely prefer all model details consolidated in one place.

Major comments:

** Figure 1 feels disjointed and lacks coordination with the main text. The authors use Fig. 1A-B to discuss population size heterogeneity (mother vs. daughter, generations), while Fig. 1C-D presents experimental data on ethanol production and respiro-fermentation. It might make more sense to integrate Fig. 1A-B with the modeling in Fig. 2, whereas Fig. 1C-D could be moved to the Supporting Information or potentially combined with Fig. 3, which fits the model to size distributions across different glucose and ethanol concentrations.

** The predictive model offers the advantage of providing deeper insights. A key result of this manuscript is that cell size scales with growth rate during fermentative growth but not during respiratory growth. I couldn't find a straightforward explanation or modeling insight for this phenomenon. Could you please elaborate on the underlying reason for this key difference in cell size scaling with growth rate between fermentation and respiration in your manuscript.

Minor comments:

** The model is an updated version of iMeGroCyc in Ref. 39. Why not call it iMeGroCyc 2.0?

** The justification and details of the sensitivity analysis for Figure 2D are unclear to me. The criteria seem arbitrary (for example, why separate parameter variation into positive and negative classes?), and the explanations are vague (such as how much the parameters were varied). Could you please clarify? One suggestion might be to create a stand-alone main figure that combines Fig. S3 (an example of T_2 sensitivity analysis) with Fig. 2D (sensitivity analysis results for all parameters).

** I am not sure if the model currently supports glucose and ethanol as dynamic variables. My assumption is that it does not, and that both experimental work and modeling are carried out under conditions of logarithmic growth where nutrients remain mostly constant. For future models, it would be valuable to incorporate factors like storage carbohydrates, secreted metabolites, and the interplay between the cell cycle and metabolism. This could provide deeper and testable insights into the origin of yeast metabolic rhythms observed in respiration and fermentation under specific bioreactor conditions.

** Typo at bottom of page 11. "As shown in Fig. 5 ..." should be "As shown in Fig. 4 ..."

**Have the authors made all data and (if applicable) computational code underlying the findings in their manuscript fully available?**

Reviewer #1: None

Reviewer #2: Yes

Reviewer #3: Yes

PLOS authors have the option to publish the peer review history of their article (what does this mean? ). If published, this will include your full peer review and any attached files.

**Do you want your identity to be public for this peer review?** For information about this choice, including consent withdrawal, please see our Privacy Policy .

Reviewer #1: No

Reviewer #2: No

Reviewer #3: No
---

## [Decision Letter · Decision Letter 1]

PCOMPBIOL-D-24-01160R1

A modular model integrating metabolism, growth, and cell cycle predicts that fermentation is required to modulate cell size in yeast populations.

PLOS Computational Biology

Dear Dr. Vanoni,

Thank you for submitting your manuscript to PLOS Computational Biology. After careful consideration, we feel that it has merit but does not fully meet PLOS Computational Biology's publication criteria as it currently stands. Therefore, we invite you to submit a revised version of the manuscript that addresses the points raised during the review process.

Please submit your revised manuscript within 60 days Apr 30 2025 11:59PM. If you will need more time than this to complete your revisions, please reply to this message or contact the journal office at ploscompbiol@plos.org. Please include the following items when submitting your revised manuscript:

We look forward to receiving your revised manuscript.

Kind regards,

Stacey D. Finley, Ph.D.

Section Editor

PLOS Computational Biology

**Additional Editor Comments:**

The reviewers appreciate the work put into improving the manuscript. However, significant issues remain - clearly showing the novel contribution of this work; providing literature references that give more recent context of this work, and improving the parameter estimation approach.

**Journal Requirements:**

1) We have noticed that you have uploaded Supporting Information files, but you have not included a list of legends. Please add a full list of legends for your Supporting Information files after the references list.

2) Please ensure that the funders and grant numbers match between the Financial Disclosure field and the Funding Information tab in your submission form. Note that the funders must be provided in the same order in both places as well.

**Reviewers' comments:**

Reviewer's Responses to Questions

**Comments to the Authors:**

Reviewer #1: Dear Authors,

While I acknowledge the additional work you have incorporated, I must express that my core concerns remain unresolved. It is still unclear what the primary aim of this study is and what its main findings are. While the manuscript builds upon your previous iMeGroCy model, which is not inherently an issue, it does not introduce new results or insights that significantly go beyond the capabilities of the earlier model. Specifically, while iMeGroCy-2 incorporates the ability to model growth on ethanol and adds a population layer, the fundamental question remains: what novel understanding does this model provide that was not accessible with the previous version? The manuscript should more clearly articulate its core contribution beyond these modifications.

My main concerns are:

1. The updated iMeGroCy-2 model extends the previous model by incorporating ethanol metabolism and a population layer. However, the manuscript struggles to demonstrate how these additions lead to significant new biological insights. For example, the model's prediction that fermentation is required for cell size modulation is interesting, but a more in-depth exploration of the underlying molecular mechanisms would strengthen the manuscript's impact. The model's ability to reproduce experimental data, while important for validation, does not, in itself, constitute a novel finding.

2. The manuscript heavily relies on outdated references, with minimal integration of recent and impactful studies that are highly relevant to the field of whole cell modeling. This not only limits its relevance but also fails to adequately situate the research within the current advancements in the field. The discussion section of the manuscript could be strengthened by a more thorough survey of the recent literature. This is particularly relevant given the increasing sophistication of whole-cell models and the integration of omics data.

3. While the model's parameter fitting process is now detailed in the supplementary material, it still relies on a trial-and-error approach with a very limited tuning set of parameters. The model’s robustness is analyzed with a sensitivity analysis. However, more justification is required to address the following: (a) How do you ensure the parameters you selected are the most biologically relevant? (b) How do you address the possibility that other parameter sets could also reproduce the same experimental data? (c) Are there other validation strategies beyond those reported, for assessing the model's biological plausibility?

4. While the model can simulate different growth conditions and mutants, the manuscript does not adequately explore the underlying molecular mechanisms. The analysis of mutants, for example, is limited to alterations in a few catalytic parameters, and a more comprehensive approach to understanding the molecular and genetic underpinnings of the observed phenotypes would make a stronger contribution. The integration of molecular data, for instance, the parameters for Whi5 mutants, could also be a model for the analysis of other pathways.

5. While the hybrid model is presented as an advantage of iMeGroCy-2, it’s not clear what is the specific value in this approach and how it enhances the model's predictive power. The manuscript needs to better explain why it is necessary to replace coarse-grained functions with more detailed molecular sub-models and why the same results cannot be obtained with the simpler model.

Reviewer #3: This is a much improved and more readable manuscript. Congrats!

My question (R3.3) was not completely answered, and I was hoping that the authors might consider adding a sentence or two for the general reader. My question was "why do yeasts with low growth rates (which are allocating more of their proteome to respiration rather than fermentation) not show cell size variation under changing growth rate conditions"? This is a well known experimental fact, but I was hoping their model could provide a satisfying answer for why there is no cell size variation under changing low growth rates. How is low lambda / large respiratory proteome feeding into their cell cycle model (with its current parameters) to generate this population-level phenomenon? Can you predict the critical lambda at which cell size changes from flat to linear, and which cell cycle parameters set the critical lambda?

**Have the authors made all data and (if applicable) computational code underlying the findings in their manuscript fully available?**

Reviewer #1: Yes

Reviewer #3: Yes

PLOS authors have the option to publish the peer review history of their article (what does this mean? ). If published, this will include your full peer review and any attached files.

**Do you want your identity to be public for this peer review?** For information about this choice, including consent withdrawal, please see our Privacy Policy .

Reviewer #1: No

Reviewer #3: No

**Figure resubmission:**
---

## [Editor Report · Decision Letter 2]

Dear Prof. Vanoni,

We are pleased to inform you that your manuscript 'A modular model integrating metabolism, growth, and cell cycle predicts that fermentation is required to modulate cell size in yeast populations.' has been provisionally accepted for publication in PLOS Computational Biology.

Best regards,

Stacey D. Finley, Ph.D.

Section Editor

PLOS Computational Biology

---

## [Editor Report · Acceptance letter]

PCOMPBIOL-D-24-01160R2

A modular model integrating metabolism, growth, and cell cycle predicts that fermentation is required to modulate cell size in yeast populations.

Dear Dr Vanoni,

I am pleased to inform you that your manuscript has been formally accepted for publication in PLOS Computational Biology. Your manuscript is now with our production department and you will be notified of the publication date in due course.

With kind regards,

Lilla Horvath
